# How Do Transformers Learn In-Context Beyond Simple Functions? A Case Study on Learning with Representations

**Tianyu Guo**[1]  **Wei Hu**[2]  **Song Mei**[1]
**Huan Wang**[3]  **Caiming Xiong**[3]  **Silvio Savarese**[3]  **Yu Bai**[3]
[1]UC Berkeley  [2]University of Michigan  [3]Salesforce AI Research
`tianyu_guo@berkeley.edu`

## Abstract

While large language models based on the transformer architecture have demonstrated remarkable in-context learning (ICL) capabilities, understandings of such capabilities are still in an early stage, where existing theory and mechanistic understanding focus mostly on simple scenarios such as learning simple function classes. This paper takes initial steps on understanding ICL in more complex scenarios, by studying learning with *representations*. Concretely, we construct synthetic in-context learning problems with a compositional structure, where the label depends on the input through a possibly complex but *fixed* representation function (which we instantiate as multi-layer MLPs), composed with a linear function that *differs* in each instance. By construction, the optimal ICL algorithm first transforms the inputs by the representation function, and then performs linear ICL on top of the transformed dataset. We show theoretically the existence of transformers that approximately implement such algorithms with mild depth and size. Empirically, we find trained transformers consistently achieve near-optimal ICL performance in this setting, and exhibit the desired dissection where lower layers transforms the dataset and upper layers perform linear ICL. Through extensive probing and a new pasting experiment, we further reveal several mechanisms within the trained transformers, such as concrete copying behaviors on both the inputs and the representations, linear ICL capability of the upper layers alone, and a post-ICL representation selection mechanism in a harder mixture setting. These observed mechanisms align well with our theory and may shed light on how transformers perform ICL in more realistic scenarios.

## 1 Introduction

Large language models based on the transformer architecture have demonstrated remarkable in-context learning (ICL) capabilities (Brown et al., 2020), where they can solve newly encountered tasks when prompted with only a few training examples, without any parameter update to the model. Recent state-of-the-art models further achieve impressive performance in context on sophisticated real-world tasks (OpenAI, 2023; Bubeck et al., 2023; Touvron et al., 2023). Such remarkable capabilities call for better understandings, which recent work tackles from various angles (Xie et al., 2021; Chan et al., 2022; Razeghi et al., 2022; Min et al., 2022; Olsson et al., 2022; Wei et al., 2023).

A recent surge of work investigates ICL in a theoretically amenable setting where the context consists of real-valued (input, label) pairs generated from a certain function class. They find that transformers can learn many function classes in context, such as linear functions, shallow neural networks, and decision trees (Garg et al., 2022; Akyürek et al., 2022; Li et al., 2023a), and further studies provide theoretical justification on how transformers can implement and learn various learning algorithms in-context such as ridge regression (Akyürek et al., 2022), gradient descent (von Oswald et al., 2022; Dai et al., 2022; Zhang et al., 2023a; Ahn et al., 2023), algorithm selection (Bai et al., 2023), and Bayes model averaging (Zhang et al., 2023b), to name a few. Despite the progress, an insufficiency of this line is that the settings and results may not actually resemble ICL in real-world scenarios—For example, ICL in linear function classes are well understood in theory with

(a) Illustration of our setting and theory     (b) ICL risks     (c) Linear probes

Figure 1: An illustration of our setting and results. **(a)** We consider ICL problems with a fixed representation composed with changing linear functions, and we construct transformers that first compute the representations and then performs linear ICL. **(b,c)** Empirically, learned transformers can perform near-optimal ICL in this setting, and exhibit mechanisms that align with our theory (detailed setups in Section 5.1).

efficient transformer constructions (Bai et al., 2023), and transformers indeed learn them well empirically (Garg et al., 2022); however, such linear functions in the raw input may fail to capture real-world scenarios where *prior knowledge* can often aid learning.

This paper takes initial steps towards addressing this by studying ICL in the setting of *learning with representations*, a more complex and perhaps more realistic setting than existing ones. We construct synthetic ICL tasks where labels depend on inputs through a fixed representation function composed with a varying linear function. We instantiate the representation as shallow neural networks (MLPs), and consider both a supervised learning setting (with input-label pairs) and a dynamical systems setting (with inputs only) for the in-context data. Our contributions can be summarized as follows.

- Theoretically, we construct transformers that implement in-context ridge regression on the representations (which includes the Bayes-optimal algorithm) for both learning settings (Section 4). Our transformer constructions admit mild sizes, and can predict at every token using a decoder architecture, (non-trivially) generalizing existing efficient constructions that predict at the last token only using an encoder architecture.

- Empirically, using $L$-layer MLPs as representations, we find that trained small transformers consistently achieve near-optimal ICL risk in both learning settings (Section 5 & Figure 1b).

- Using linear probing techniques, we identify evidence for various mechanisms in the trained transformers. Our high-level finding is that the lower layers transforms the data by the representation and prepares it into a certain format, and the upper layers perform linear ICL on top of the transformed data (Figure 1c), with often a clear dissection between these two modules, consistent with our theory. See Figure 1a for a pictorial illustration.

- We further observe several lower-level behaviors using linear probes that align well with our (and existing) theoretical constructions, such as copying (of both the input and the representations) where which tokens are being copied are precisely identifiable (Section 5.2), and a post-ICL representation selection mechanism in a harder setting (Section 5.1.1 & Appendix E).

- We perform a new pasting experiment and find that the upper layers within the trained transformer can perform nearly-optimal linear ICL in (nearly-)isolation (Section 5.1), which provides stronger evidence that the upper module alone can be a strong linear ICL learner.

## 2 RELATED WORK

**In-context learning** The in-context learning (ICL) capabilities of pretrained transformers have gained significant attention since first demonstrated with GPT-3 (Brown et al., 2020). Subsequent empirical studies have investigated the capabilities and limitations of ICL in large language models (Liu et al., 2021; Min et al., 2021a;b; Lu et al., 2021; Zhao et al., 2021; Rubin et al., 2021; Razeghi et al., 2022; Elhage et al., 2021; Kirsch et al., 2022; Wei et al., 2023).

A line of recent work investigates why and how pretrained transformers perform ICL from a theoretical perspective (Garg et al., 2022; Li et al., 2023a; von Oswald et al., 2022; Akyürek et al., 2022; Xie et al., 2021; Bai et al., 2023; Zhang et al., 2023a;b; Ahn et al., 2023; Raventós et al., 2023). In particular, Xie et al. (2021) proposed a Bayesian inference framework explaining ICL. Garg et al. (2022) showed transformers could be trained from scratch for ICL of simple function classes. Other studies found transformers can implement ICL through in-context gradient descent

(von Oswald et al., 2022; Akyürek et al., 2022) and in-context algorithm selection (Bai et al., 2023). Zhang et al. (2023a) studied the training dynamics of a single attention layer on linear ICL tasks. Li et al. (2023b) used the ICL framework to explain chain-of-thought reasoning (Wei et al., 2022). Our work builds on and extends the work of (Garg et al., 2022; Akyürek et al., 2022; von Oswald et al., 2022; Bai et al., 2023), where we study the more challenging setting of ICL with a representation function, and also provide new efficient ICL constructions for predicting at every token using a decoder transformer, as opposed to predicting only at the last token in most of these work.

**In-weights learning versus in-context learning** Recent work has investigated when transformers learn a fixed input-label mapping versus when they perform ICL (Chan et al., 2022; Wei et al., 2023; Bietti et al., 2023). Chan et al. (2022) refer to learning a fixed input-label mapping from the pre-training data as "in-weights learning" (IWL), in contrast with ICL. Our problem setting assumes the pre-training data admits a fixed representation function, which should be learned by IWL. In this perspective, unlike these existing works where IWL and ICL are typically treated as competing mechanisms, we study a model in which IWL (computing the fixed representation by transformer weights) and ICL (learning the changing linear function in context) occur simultaneously.

**Mechanistic understanding and probing techniques** A line of work focuses on developing techniques for understanding the mechanisms of neural networks, in particular transformers (Alain & Bengio, 2016; Geiger et al., 2021; Meng et al., 2022; von Oswald et al., 2022; Akyürek et al., 2022; Wang et al., 2022; Räuker et al., 2023). We adopted the linear probing technique of (Alain & Bengio, 2016) in a token-wise fashion for interpreting the ICL mechanisms of transformers. Beyond probing, more convincing mechanistic interpretations may require advanced approaches such as causal intervention (Geiger et al., 2021; Vig et al., 2020; Wang et al., 2022); Our pasting experiment has a similar interventional flavor in that we feed input sequences (ICL instances) from another distribution directly (through a trainable embedding layer) to the upper module of a transformer.

## 3 PRELIMINARIES

**Transformers** We consider sequence-to-sequence functions applied to $N$ input vectors $\{\mathbf{h}_i\}_{i=1}^N \subset \mathbb{R}^{D_{\text{hid}}}$ in $D_{\text{hid}}$ dimensions, which we write compactly as an input matrix $\mathbf{H} = [\mathbf{h}_1, \ldots, \mathbf{h}_N] \in \mathbb{R}^{D_{\text{hid}} \times N}$, where each $\mathbf{h}_i$ is a column of $\mathbf{H}$ (also a *token*).

We use a standard $L$-layer decoder-only (autoregressive) transformer, which consists of $L$ consecutive blocks each with a masked self-attention layer (henceforth "attention layer") followed by an MLP layer. Each attention layer computes

$$\text{Attn}_{\boldsymbol{\theta}}(\mathbf{H}) := \mathbf{H} + \sum_{m=1}^M (\mathbf{V}_m \mathbf{H}) \times \overline{\sigma}\big(\text{MSK} \odot ((\mathbf{Q}_m \mathbf{H})^\top (\mathbf{K}_m \mathbf{H}))\big) \in \mathbb{R}^{D \times N},$$

where $\boldsymbol{\theta} = \{(\mathbf{Q}_m, \mathbf{K}_m, \mathbf{V}_m) \subset \mathbb{R}^{D_{\text{hid}} \times D_{\text{hid}}}\}_{m \in [M]}$ are the (query, key, value) matrices, $M$ is the number of heads, $\text{MSK} \in \mathbb{R}^{N \times N}$ is the decoder mask matrix with $\text{MSK}_{ij} = 1\{i \leq j\}$, and $\overline{\sigma}$ is the activation function which is typically chosen as the (column-wise) softmax: $[\overline{\sigma}(\mathbf{A})]_{:,j} = \text{softmax}(\mathbf{a}_j) \in \mathbb{R}^N$ for $\mathbf{A} = [\mathbf{a}_1, \ldots, \mathbf{a}_N] \in \mathbb{R}^{N \times N}$. Each MLP layer computes

$$\text{MLP}_{\mathbf{W}_1, \mathbf{W}_2}(\mathbf{H}) := \mathbf{H} + \mathbf{W}_2 \sigma(\mathbf{W}_1 \mathbf{H}),$$

where $\mathbf{W}_{\{1,2\}} \in \mathbb{R}^{D_{\text{hid}} \times D_{\text{hid}}}$ are the weight matrices, and $\sigma(t) = \max\{t, 0\}$ is the ReLU activation. We use TF to denote a transformer, and typically use $\widetilde{\mathbf{H}} = \text{TF}(\mathbf{H})$ to denote its output on $\mathbf{H}$.

**In-context learning** We consider in-context learning (ICL) on regression problems, where each ICL instance is specified by a dataset $\mathcal{D} = \{(\mathbf{x}_i, y_i)\}_{i \in [N]} \overset{\text{iid}}{\sim} \mathsf{P}$, with $(\mathbf{x}_i, y_i) \in \mathbb{R}^d \times \mathbb{R}$, and the model is required to accurately predict $y_i$ given all past observations $\mathcal{D}_{i-1} := \{(\mathbf{x}_j, y_j)\}_{j \leq i-1}$ and the test input $\mathbf{x}_i$. Each instance $\mathcal{D} = \mathcal{D}^{(j)}$ is drawn from a different data distribution $\mathsf{P} = \mathsf{P}^{(j)}$. Accurate prediction requires learning $\mathsf{P}$ in-context from the past observations $\mathcal{D}_{i-1}$ (i.e. the context); merely memorizing any fixed $\mathsf{P}^{(j)}$ is not enough. This is a main challenge of in-context learning.

We consider using transformers to do ICL, where we feed a sequence of length $2N$ into the transformer TF using the following input format:

$$\mathbf{H} = [\mathbf{h}_1, \ldots, \mathbf{h}_{2N}] = \begin{bmatrix} \mathbf{x}_1 & \mathbf{0} & \ldots & \mathbf{x}_N & \mathbf{0} \\ 0 & y_1 & \ldots & 0 & y_N \\ \mathbf{p}_1^x & \mathbf{p}_1^y & \cdots & \mathbf{p}_N^x & \mathbf{p}_N^y \end{bmatrix} \in \mathbb{R}^{D_{\mathrm{hid}} \times 2N}, \tag{1}$$

where $\mathbf{p}_i^x, \mathbf{p}_i^y \in \mathbb{R}^{D_{\mathrm{hid}}-d-1}$ are fixed positional encoding vectors consisting of *zero paddings*, followed by non-zero entries containing information about the position index $i$ and indicator of being an $x$-token (1 in $\mathbf{p}_i^x$, and 0 in $\mathbf{p}_i^y$); see (12) for our concrete choice. We refer to each odd token $\mathbf{h}_{2i-1}$ as as an $x$-token (also the $\mathbf{x}_i$-token), and each even token $\mathbf{h}_{2i}$ as a $y$-token (also the $y_i$-token).

After obtaining the transformer output $\widetilde{\mathbf{H}} = \mathrm{TF}(\mathbf{H})$, for every index $i \in [N]$, we extract the prediction $\widehat{y}_i$ from the output token at position $\mathbf{x}_i$: $\widehat{y}_i := (\widetilde{\mathbf{h}}_i^x)_{d+1}$.[1] Feeding input (1) into the transformer simultaneously computes $\widehat{y}_i \leftarrow \mathrm{TF}(\mathbf{x}_1, y_1, \ldots, \mathbf{x}_{i-1}, y_{i-1}, \mathbf{x}_i)$ for all $i \in [N]$. Denote the parameters of transformers as $\boldsymbol{\theta}$.

**In addition** to the above setting, we also consider a *dynamical system* setting with $\mathcal{D} = \{\mathbf{x}_i\}_{i \in [N]}$ where the transformer predicts $\widehat{\mathbf{x}}_i$ from the preceding inputs $\mathbf{x}_{\leq i-1}$. See Section 4.2 for details.

# 4 IN-CONTEXT LEARNING WITH REPRESENTATIONS

## 4.1 SUPERVISED LEARNING WITH REPRESENTATION

We begin by considering ICL on regression problems with representation, where labels depend on the input through linear functions of a fixed representation function. Formally, let $\Phi^\star : \mathbb{R}^d \to \mathbb{R}^D$ be a fixed representation function. We generate each in-context data distribution $\mathsf{P} = \mathsf{P}_{\mathbf{w}}$ by sampling a linear function $\mathbf{w} \sim \mathsf{N}(\mathbf{0}, \tau^2 \mathbf{I}_D)$ from a Gaussian prior, and then generate the ICL instance $\mathcal{D} = \{(\mathbf{x}_i, y_i)\}_{i \in [N]} \sim \mathsf{P}_{\mathbf{w}}$ by a linear model on $\Phi^\star$ with coefficient $\mathbf{w}$ and noise level $\sigma > 0$:

$$y_i = \langle \mathbf{w}, \Phi^\star(\mathbf{x}_i) \rangle + \sigma z_i, \quad \mathbf{x}_i \overset{\text{iid}}{\sim} \mathsf{P}_x, \quad z_i \overset{\text{iid}}{\sim} \mathsf{N}(0, 1), \quad i \in [N]. \tag{2}$$

Note that all $\mathcal{D}$'s share the same representation $\Phi^\star$, but each admits a unique linear function $\mathbf{w}$.

The representation function $\Phi^\star$ can in principle be chosen arbitrarily. As a canonical and flexible choice for both our theory and experiments, we choose $\Phi^\star$ to be a standard $L$-layer MLP:

$$\Phi^\star(\mathbf{x}) = \sigma^\star\big(\mathbf{B}_L^\star \sigma^\star\big(\mathbf{B}_{L-1}^\star \cdots \sigma^\star(\mathbf{B}_1^\star \mathbf{x}) \cdots\big)\big), \quad \mathbf{B}_1^\star \in \mathbb{R}^{D \times d}, \quad (\mathbf{B}_\ell^\star)_{\ell=2}^L \subset \mathbb{R}^{D \times D} \tag{3}$$

where $D$ is the hidden and output dimension, and $\sigma^\star$ is the activation function (applied entry-wise) which we choose to be the leaky ReLU $\sigma^\star(t) = \sigma_\rho(t) := \max\{t, \rho t\}$ with slope $\rho \in (0, 1)$.

**Theory** As $\Phi^\star$ is fixed and the $\mathbf{w}$ is changing in model (2), by construction, a good ICL algorithm should *compute* the representations $\{\Phi^\star(\mathbf{x}_i)\}_i$ and perform linear ICL on the transformed dataset $\{(\Phi^\star(\mathbf{x}_i), y_i)\}_i$ to learn $\mathbf{w}$. We consider the following class of $\Phi^\star$-*ridge* estimators:

$$\widehat{\mathbf{w}}_i^{\Phi^\star, \lambda} := \arg\min_{\mathbf{w} \in \mathbb{R}^d} \frac{1}{2(i-1)} \sum_{j=1}^{i-1} \left( \langle \mathbf{w}, \Phi^\star(\mathbf{x}_j) \rangle - y_j \right)^2 + \frac{\lambda}{2} \|\mathbf{w}\|_2^2, \quad (\Phi^\star\text{-Ridge})$$

and we understand $\widehat{\mathbf{w}}_1^{\Phi^\star, \lambda} := \mathbf{0}$. In words, $\widehat{\mathbf{w}}_i^{\Phi^\star, \lambda}$ performs ridge regression on the transformed dataset $\{\Phi(\mathbf{x}_j), y_j\}_{j \leq i-1}$ for all $i \in [N]$. By standard calculations, the Bayes-optimal predictor[2] for $y_i$ given $(\mathcal{D}_{i-1}, \mathbf{x}_i)$ is exactly the ridge predictor $\widehat{y}_i^{\Phi^\star, \lambda} := \langle \widehat{\mathbf{w}}_i^{\Phi^\star, \lambda}, \Phi^\star(\mathbf{x}_i) \rangle$ at $\lambda = \sigma^2/\tau^2$.

We show that there exists a transformer that can approximately implement ($\Phi^\star$-Ridge) in-context at every token $i \in [N]$. The proof can be found in Appendix B.

**Theorem 1** (Transformer can implement $\Phi^\star$-Ridge). *For any representation function $\Phi^\star$ of form (3), any $\lambda > 0$, $B_\Phi, B_w, B_y > 0$, $\varepsilon < B_\Phi B_w / 2$, letting $\kappa := 1 + B_\Phi^2 / \lambda$, there exists a transformer TF with $L + \mathcal{O}(\kappa \log(B_\Phi B_w / \varepsilon))$ layers, 5 heads, $D_{\mathrm{hid}} = 2D + d + 10$ such that the following holds.*

*For any dataset $\mathcal{D}$ such that $\|\Phi^\star(\mathbf{x}_i)\|_2 \leq B_\Phi$, $|y_i| \leq B_y$ and the corresponding input $\mathbf{H} \in \mathbb{R}^{D_{\mathrm{hid}} \times 2N}$ of format (1), we have*

---

[1] There is no information leakage, as the "prefix" property of decoder transformers $\widetilde{\mathbf{h}}_i^x = \widetilde{\mathbf{h}}_{2i-1} = [\mathrm{TF}(\mathbf{H}_{:,1:(2i-1)})]_{2i-1}$ ensures that $\widetilde{\mathbf{h}}_i^x$ (and thus $\widehat{y}_i$) only depends on $(\mathcal{D}_{i-1}, \mathbf{x}_i)$.

[2] The predictor $\widehat{y}_i = \widehat{y}_i(\mathcal{D}_{i-1}, \mathbf{x}_i)$ that minimizes the posterior square loss $\mathbb{E}[\frac{1}{2}(\widehat{y}_i - y_i)^2 | \mathcal{D}_{i-1}, \mathbf{x}_i]$.

(a) *The first $(L+2)$ layers of* TF *transforms $\mathbf{x}_i$ to the representation $\Phi^\star(\mathbf{x}_i)$ at each $x$ token, and copies them into the succeeding $y$ token:*

$$\mathrm{TF}^{(1:L+2)}(\mathbf{H}) = \begin{bmatrix} \Phi^\star(\mathbf{x}_1) & \Phi^\star(\mathbf{x}_1) & \dots & \Phi^\star(\mathbf{x}_N) & \Phi^\star(\mathbf{x}_N) \\ 0 & y_1 & \dots & 0 & y_N \\ \widetilde{\mathbf{p}}_1^x & \widetilde{\mathbf{p}}_1^y & \dots & \widetilde{\mathbf{p}}_N^x & \widetilde{\mathbf{p}}_N^y \end{bmatrix}, \tag{4}$$

*where $\widetilde{\mathbf{p}}_i^x, \widetilde{\mathbf{p}}_i^y$ only differ from $\mathbf{p}_i^x, \mathbf{p}_i^y$ in the dimension of the zero paddings.*

(b) *For every index $i \in [N]$, the transformer output $\widetilde{\mathbf{H}} = \mathrm{TF}(\mathbf{H})$ contains prediction $\widehat{y}_i := [\widetilde{\mathbf{h}}_{2i-1}]_{D+1}$ that is close to the ($\Phi^\star$-Ridge) predictor: $|\widehat{y}_i - \langle \Phi^\star(\mathbf{x}_i), \widehat{\mathbf{w}}_i^{\Phi^\star,\lambda}\rangle| \le \varepsilon$.*

The transformer construction in Theorem 1 consists of two "modules": The lower layers computes the representations and prepares the transformed dataset $\{(\Phi^\star(\mathbf{x}_i), y_i)\}_i$ into form (4). In particular, each $\Phi^\star(\mathbf{x}_i)$ appears both in the $i$-th $x$-token and is also copied into the succeeding $y$ token. The upper layers perform linear ICL (ridge regression) on top of the transformed dataset. We will test whether such mechanisms align with trained transformers in reality in our experiments (Section 5.1).

**Proof techniques** The proof of Theorem 1 builds upon (1) implementing the MLP $\Phi^\star$ by transformers (Lemma B.3), and (2) an efficient construction of in-context ridge regression (Theorem B.5), which to our knowledge is the first efficient construction for predicting *at every token* using decoder transformers. The latter requires several new construction techniques such as a copying layer (Lemma B.1), and an efficient implementation of $N$ parallel in-context gradient descent algorithms at all tokens simultaneously using a decoder transformer (Proposition B.4). These extend the related constructions of von Oswald et al. (2022); Bai et al. (2023) who only consider predicting at the last token using encoder transformer, and could be of independent interest.

In addition, the bounds on the number of layers, heads, and $D_{\mathrm{hid}}$ in Theorem 1 can imply a sample complexity guarantee for (pre-)training: A transformer with $\widetilde{\varepsilon}$-excess risk (on the same ICL instance distribution) over the one constructed in Theorem 1 can be found in $\widetilde{\mathcal{O}}\big((L+\kappa)^2(D+d)^2\widetilde{\varepsilon}^{-2}\big)$ training instances, by the generalization analysis of Bai et al. (2023, Theorem 20). We remark that the constructions in Theorem 1 & 2 choose $\overline{\sigma}$ as the normalized ReLU instead of softmax, following (Bai et al., 2023) and in resonance with recent empirical studies (Wortsman et al., 2023).

## 4.2 Dynamical System with Representation

As a variant of model (2), we additionally consider a (nonlinear) dynamical system setting with data $\mathcal{D} = (\mathbf{x}_1, \dots, \mathbf{x}_N)$, where each $\mathbf{x}_{i+1}$ depends on the $k$ preceding inputs $[\mathbf{x}_{i-k+1}; \dots; \mathbf{x}_i]$ for some $k \ge 1$ through a linear function on top of a fixed representation function $\Phi^\star$. Compared to the supervised learning setting in Section 4.1, this setting better resembles some aspects of natural language, where the next token in general depends on several preceding tokens.

Formally, let $k \ge 1$ denote the number of input tokens that the next token depends on, and $\Phi^\star : \mathbb{R}^{kd} \to \mathbb{R}^D$ denotes a representation function. Each ICL instance $\mathcal{D} = \{\mathbf{x}_i\}_{i\in[N]}$ is generated as follows: First sample $\mathsf{P} = \mathsf{P}_{\mathbf{W}}$ where $\mathbf{W} \in \mathbb{R}^{D\times d}$ is sampled from a Gaussian prior: $W_{ij} \overset{\mathrm{iid}}{\sim} \mathsf{N}(0, \tau^2)$. Then sample the initial input $\mathbf{x}_1 \sim \mathsf{P}_x$ and let

$$\mathbf{x}_{i+1} = \mathbf{W}^\top \Phi^\star([\mathbf{x}_{i-k+1}; \dots; \mathbf{x}_i]) + \sigma \mathbf{z}_i, \quad \mathbf{z}_i \overset{\mathrm{iid}}{\sim} \mathsf{N}(\mathbf{0}, \mathbf{I}_d), \quad i \in [N-1], \tag{5}$$

where we understand $\mathbf{x}_j := \mathbf{0}_d$ for $j \le 0$. We choose $\Phi^\star$ to be the same $L$-layer MLP as in (3), except that the first weight matrix has size $\mathbf{B}_1^\star \in \mathbb{R}^{D\times kd}$ to be consistent with the dimension of the augmented input $\overline{\mathbf{x}}_i := [\mathbf{x}_{i-k+1}; \dots; \mathbf{x}_i]$. We remark that (5) substantially generalizes the setting of Li et al. (2023a) which only considers *linear* dynamical systems (equivalent to $\Phi^\star \equiv \mathrm{id}$), a task arguably much easier for transformers to learn in context.

As $\mathbf{x}_i$ acts as both inputs and labels in model (5), we use the following input format for transformers:

$$\mathbf{H} := \begin{bmatrix} \mathbf{x}_1 & \dots & \mathbf{x}_N \\ \mathbf{p}_1 & \dots & \mathbf{p}_N \end{bmatrix} \in \mathbb{R}^{D_{\mathrm{hid}} \times N}, \tag{6}$$

where $\mathbf{p}_i := [\mathbf{0}_{D_{\mathrm{hid}}-d-4}; 1; i; i^2; i^3]$, and we extract prediction $\widehat{\mathbf{x}}_{i+1}$ from the $i$-th output token.

**Theory** Similar as above, we consider the ridge predictor for the dynamical system setting

$$\widehat{\mathbf{W}}_i^{\Phi^\star,\lambda} := \arg\min_{\mathbf{W}\in\mathbb{R}^{D\times d}} \frac{1}{2(i-1)} \sum_{j=1}^{i-1} \left\| \mathbf{W}^\top \Phi^\star(\overline{\mathbf{x}}_j) - \mathbf{x}_{j+1} \right\|_2^2 + \frac{\lambda}{2} \|\mathbf{W}\|_{\mathsf{Fr}}^2. \quad (\Phi^\star\text{-Ridge-Dyn})$$

We understand $\widehat{\mathbf{W}}_0^{\Phi^\star,\lambda} := \mathbf{0}_{D\times d}$, and let $\|\mathbf{W}\|_{2,\infty} := \max_{j\in[d]} \|\mathbf{W}_{:,j}\|_2$ for any $\mathbf{W} \in \mathbb{R}^{D\times d}$. Again, ($\Phi^\star$-Ridge-Dyn) gives the Bayes-optimal predictor $(\widehat{\mathbf{W}}_i^{\Phi^\star,\lambda})^\top \Phi^\star(\overline{\mathbf{x}}_i)$ at $\lambda = \sigma^2/\tau^2$.

The following result shows that ($\Phi^\star$-Ridge-Dyn) can also be implemented efficiently by a transformer. The proof can be found in Appendix C.2.

**Theorem 2** (Transformer can implement $\Phi^\star$-Ridge for dynamical system)**.** *For the dynamical system setting where the $L$-layer representation function $\Phi^\star : \mathbb{R}^{kd} \to \mathbb{R}^D$ takes form (3), but otherwise same settings as Theorem 1, there exists a transformer* TF *with $L+2+\mathcal{O}(\kappa \log(B_\Phi B_w/\varepsilon))$ layers, $\max\{3d,5\}$ heads, and $D_{\mathrm{hid}} = \max\{2(k+1),D\}d+3(D+d)+5$ such that the following holds.*

*For any dataset $\mathcal{D}$ such that $\|\Phi^\star(\overline{\mathbf{x}}_i)\|_2 \le B_\Phi$, $\|\mathbf{x}_i\|_\infty \le B_y$, and $\|\widehat{\mathbf{W}}_i^{\Phi^\star,\lambda}\|_{2,\infty} \le B_w/2$ (cf. ($\Phi^\star$-Ridge-Dyn)) for all $i \in [N]$, and corresponding input $\mathbf{H} \in \mathbb{R}^{D_{\mathrm{hid}}\times N}$ of format (6), we have*

(a) *The first transformer layer copies the $k$ previous inputs into the current token, and computes the first layer $\{\sigma_\rho(\mathbf{B}_1^\star\overline{\mathbf{x}}_i)\}_{i\in[N]}$ within $\Phi^\star$:*

$$\mathrm{Attn}^{(1)}(\mathbf{H}) = \begin{bmatrix} \overline{\mathbf{x}}_1 & \cdots & \overline{\mathbf{x}}_N \\ \overline{\mathbf{p}}_1 & \cdots & \overline{\mathbf{p}}_N \end{bmatrix} = \begin{bmatrix} \mathbf{x}_{1-k+1} & \cdots & \mathbf{x}_{N-k+1} \\ | & & | \\ \mathbf{x}_1 & \cdots & \mathbf{x}_N \\ \overline{\mathbf{p}}_1 & \cdots & \overline{\mathbf{p}}_N \end{bmatrix}; \quad (7)$$

$$\mathrm{TF}^{(1)}(\mathbf{H}) = \mathrm{MLP}^{(1)}\left(\mathrm{Attn}^{(1)}(\mathbf{H})\right) = \begin{bmatrix} \sigma_\rho(\mathbf{B}_1^\star\overline{\mathbf{x}}_1) & \cdots & \sigma_\rho(\mathbf{B}_1^\star\overline{\mathbf{x}}_N) \\ \mathbf{x}_1 & \cdots & \mathbf{x}_N \\ \overline{\mathbf{p}}_1' & \cdots & \overline{\mathbf{p}}_N' \end{bmatrix}. \quad (8)$$

(b) *The first $(L+1)$ layers of* TF *transforms each $\mathbf{x}_i$ to $\Phi^\star(\overline{\mathbf{x}}_i)$, and copies the preceding representation $\Phi^\star(\overline{\mathbf{x}}_{i-1})$ onto the same token to form the (input, label) pair $(\Phi^\star(\overline{\mathbf{x}}_{i-1}), \mathbf{x}_i)$:*

$$\mathrm{TF}^{(1:L+1)}(\mathbf{H}) = \begin{bmatrix} \Phi^\star(\overline{\mathbf{x}}_1) & \Phi^\star(\overline{\mathbf{x}}_2) & \cdots & \Phi^\star(\overline{\mathbf{x}}_N) \\ \mathbf{0}_d & \mathbf{0}_d & \cdots & \mathbf{0}_d \\ \mathbf{0}_D & \Phi^\star(\overline{\mathbf{x}}_1) & \cdots & \Phi^\star(\overline{\mathbf{x}}_{N-1}) \\ \mathbf{x}_1 & \mathbf{x}_2 & \cdots & \mathbf{x}_N \\ \widetilde{\mathbf{p}}_1 & \widetilde{\mathbf{p}}_2 & \cdots & \widetilde{\mathbf{p}}_N \end{bmatrix}. \quad (9)$$

*Above, $\overline{\mathbf{p}}_i, \overline{\mathbf{p}}_i', \widetilde{\mathbf{p}}_i$ only differs from $\mathbf{p}_i$ in the dimension of the zero paddings.*

(c) *For every index $i \in [N]$, the transformer output $\widetilde{\mathbf{H}} = \mathrm{TF}(\mathbf{H})$ contains prediction $\widehat{\mathbf{x}}_{i+1} := [\widetilde{\mathbf{h}}_i]_{1:d}$ that is close to the ($\Phi^\star$-Ridge-Dyn) predictor: $\|\widehat{\mathbf{x}}_{i+1} - (\widehat{\mathbf{W}}_i^{\Phi^\star,\lambda})^\top \Phi^\star(\overline{\mathbf{x}}_i)\|_\infty \le \varepsilon$.*

To our best knowledge, Theorem 2 provides the first transformer construction for learning nonlinear dynamical systems in context. Similar as for Theorem 1, the bounds on the transformer size here imply guarantees $\widetilde{\varepsilon}$ excess risk within $\widetilde{\mathcal{O}}\big((L+\kappa)^2((k+D)d)^2\widetilde{\varepsilon}^{-2}\big)$ (pre-)training instances.

In terms of the mechanisms, compared with Theorem 1, the main differences in Theorem 2 are (1) the additional copying step (7) within the first layer, where the previous $(k-1)$ tokens $[\mathbf{x}_{i-k+1}; \ldots; \mathbf{x}_{i-1}]$ are copied onto the $\mathbf{x}_i$ token, to prepare for computing of $\Phi^\star(\overline{\mathbf{x}}_i)$; (2) the intermediate output (9), where relevant information (for preparing for linear ICL) has form $[\Phi^\star(\overline{\mathbf{x}}_{i-1}); \mathbf{x}_i; \Phi^\star(\overline{\mathbf{x}}_i)]$ and is gathered in the $\mathbf{x}$-tokens, different from (4) where the relevant information is $[\Phi^\star(\mathbf{x}_i); y_i]$, gathered in the $y$-token. We will test these in our experiments (Section 5.2).

## 5 EXPERIMENTS

We now empirically investigate trained transformers under the two settings considered in Section 4.1 & 4.2. In both cases, we choose the representation function $\Phi^\star$ to be a normalized version of the $L$-layer MLP (3): $\Phi^\star(\mathbf{x}) := \widetilde{\Phi}^\star(\mathbf{x})/\|\widetilde{\Phi}^\star(\mathbf{x})\|_2$, where $\widetilde{\Phi}^\star$ takes form (3), with weight matrices $(\mathbf{B}_i^\star)_{i\in[L]}$ sampled as random (column/row-orthogonal) matrices and held fixed in each experiment,

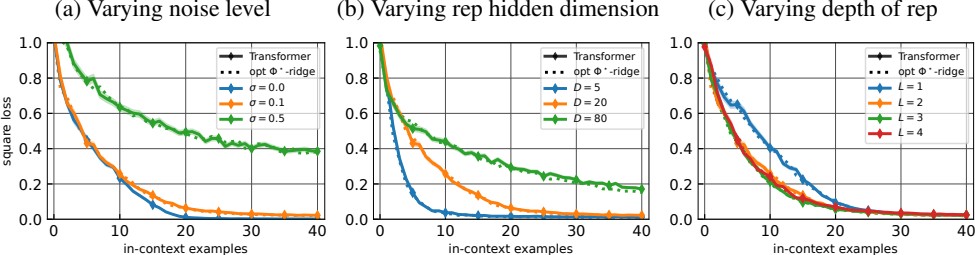

Figure 2: Test ICL risk for learning with representations. Each plot modifies a single problem parameter from the base setting $(L, D, \sigma) = (2, 20, 0.1)$. Dotted lines plot the Bayes-optimal risks for each setting respectively.

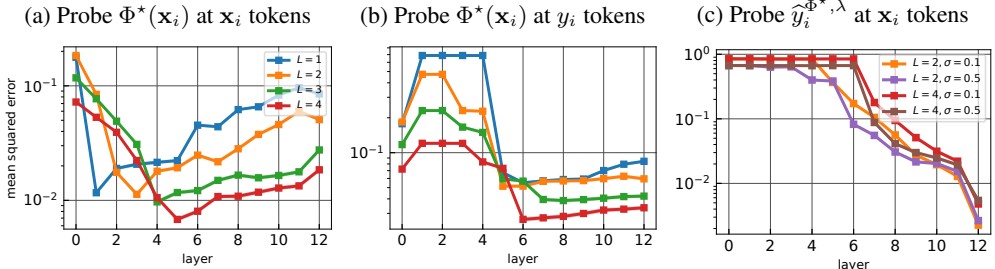

Figure 3: Probing errors for the learning with representation setting. Each setting modifies one or two problem parameters from the base setting $(L, D, \sigma) = (2, 20, 0.1)$. Note that the orange curve corresponds to the same setting (and thus the same transformer) across (a,b,c), as well as the red curve.

and slope $\rho = 0.01$. We test $L \in \{1, 2, 3, 4\}$, hidden dimension $D \in \{5, 20, 80\}$, and noise level $\sigma \in \{0, 0.1, 0.5\}$. All experiments use $\mathsf{P}_x = \mathsf{N}(\mathbf{0}, \mathbf{I}_d)$, $\tau^2 = 1$, $d = 20$, and $N = 41$.

We use a small architecture within the GPT-2 family with 12 layers, 8 heads, and $D_{\mathrm{hid}} = 256$, following (Garg et al., 2022; Li et al., 2023a; Bai et al., 2023). The (pre)-training objective for the transformer (for the supervised learning setting) is the average prediction risk at all tokens:

$$\min_{\boldsymbol{\theta}} \mathbb{E}_{\mathbf{w}, \mathcal{D} \sim \mathsf{P}_{\mathbf{w}}} \left[ \frac{1}{2N} \sum_{i=1}^{N} \left( \widehat{y}_{\boldsymbol{\theta}, i}(\mathcal{D}_{i-1}, \mathbf{x}_i) - y_i \right)^2 \right], \tag{10}$$

where $\widehat{y}_{\boldsymbol{\theta}, i}$ is extracted from the $(2i - 1)$-th output token of $\mathrm{TF}_{\boldsymbol{\theta}}(\mathbf{H})$ (cf. Section 3). The objective for the dynamical system setting is defined similarly. Additional experimental details can be found in Appendix D, and ablation studies (e.g. along the training trajectory; cf. Figure 9) in Appendix F.

## 5.1 SUPERVISED LEARNING WITH REPRESENTATION

We first test ICL with supervised learning data as in Section 4.1, where for each configuration of $(L, D, \sigma)$ (which induces a $\Phi^\star$) we train a transformer on ICL data distribution (2) and evaluate ICL on the same distribution. Note that Figure 1c & 1b plots the results for $(L, D, \sigma) = (2, 20, 0.1)$.

**ICL performance** Figure 2 reports the test risk across various settings, where we observe that trained transformers can consistently match the Bayes-optimal ridge predictor. This extends existing results which show that linear functions (without a representation) can be learned near-optimally in-context by transformers (Garg et al., 2022; Akyürek et al., 2022), adding our model (2) to this list of (empirically) nearly-optimally learnable function classes. Among the complexity measures $(L, D, \sigma)$, observe that the noise level $\sigma$ and hidden dimension $D$ of the representation (Figure 2a & 2b) appears to have a larger effect on the (nearly Bayes-optimal) risk than the depth $L$ (Figure 2c).

**Mechanisms via linear probing** We conduct probing experiments to further understand the mechanisms of the trained transformers. In accordance with the theoretical construction in Theorem 1, our main question here is: Does the trained transformer perform the following in order:

1. Computes $\Phi^\star(\mathbf{x}_i)$ at $x_i$ tokens;
2. Copies them onto the following $y_i$ token and obtains dataset $\{\Phi^\star(\mathbf{x}_i), y_i\}_i$ in the form of (4);
3. Performs linear ICL on top of $\{\Phi^\star(\mathbf{x}_i), y_i\}_i$?

(a) Illustration of the pasting experiment      (b) Linear ICL in TF_upper via pasting

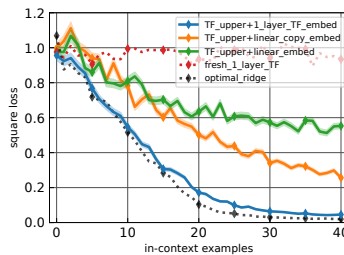

Figure 4: **(a)** Illustration of our pasting experiment, which examines the linear ICL capability of the upper module of a trained transformer. **(b)** Pasting results for the upper module of a trained transformer in setting $(L, D, \sigma) = (3, 20, 0.1)$. "TF_upper+..." correspond to feeding the upper module of trained transformer with different embeddings. It achieves nearly optimal linear ICL risk (in 20 dimension with noise 0.1), using a 1-layer transformer embedding, and also non-trivial performance using linear and linear copy embeddings.

While such internal mechanisms are in general difficult to quantify exactly, we adapt the *linear probing* (Alain & Bengio, 2016) technique to the transformer setting to identify evidence. Linear probing allows us to test whether intermediate layer outputs (tokens) $\{\mathbf{h}_i^{x,(\ell)}\}_{\ell \in [12]}$ ($\ell$ denotes the layer) and $\{\mathbf{h}_i^{y,(\ell)}\}_{\ell \in [12]}$ "contains" various quantities of interest, by linearly regressing these quantities (as the y) on the intermediate tokens (as the x), pooled over the token index $i \in [N]$. For example, regressing $\Phi^\star(\mathbf{x}_i)$ on $\mathbf{h}_i^{x,(\ell)}$ tests whether the $\mathbf{x}_i$ token after the $\ell$-th layer "contains" $\Phi^\star(\mathbf{x}_i)$, where a smaller error indicates a better containment. See Appendix D.1 for further setups of linear probing.

Figure 3 reports the errors of three linear probes across all 12 layers: The representation $\Phi^\star(\mathbf{x}_i)$ in the $\mathbf{x}_i$ tokens and $y_i$ tokens, and the optimal ridge prediction $\widehat{y}_i^{\Phi^\star, \lambda}$ in the $\mathbf{x}_i$ tokens. Observe that the probing errors for the representation decrease through lower layers and then increase through upper layers (Figure 3a & 3b), whereas probing errors for the ridge prediction monotonically decrease through the layers (Figure 3c), aligning with our construction that the transformer first computes the representations and then performs ICL on top of the representation. Also note that deeper representations take more layers to compute (Figure 3a). Further, the representation shows up later in the $y$-tokens (layers 5-6) than in the $x$-tokens (layers 1,3,4,5), consistent with the copying mechanism, albeit the copying appears to be lossy (probe errors are higher at $y$-tokens).

Finally, observe that the separation between the lower and upper modules seems to be strong in certain runs—For example, the red transformer ($L = 4, \sigma = 0.1$) computes the representation at layer 5, copies them onto $y$-tokens at layer 6, and starts to perform iterative ICL from layer 7, which aligns fairly well with our theoretical constructions at a high level.

**Investigating upper module via pasting** To further investigate upper module, we test whether it is indeed a strong ICL learner *on its own* without relying on the lower module, which would provide stronger evidence that the upper module performs linear ICL. However, a key challenge here is that it is unclear how to feed raw inputs directly into the upper module, as they supposedly only admit input formats emitted from the lower module—the part we wanted to exclude in the first place.

We address this by conducting a *pasting* experiment, where we feed $D$-dimensional *linear ICL problems* ($y_i' = \langle \mathbf{w}', \mathbf{x}_i' \rangle$ *without* a representation) with input format (1) directly to the upper module of the transformer trained on representation $\Phi^\star$, by adding a *trainable embedding layer* in between; see Figure 4a for an illustration of the pasting approach. This trainable embedding layer itself needs to be shallow without much ICL power—we test the following three choices: (1) *Linear* embedding: $\overline{\mathbf{h}}_i^x = \mathbf{W}[\mathbf{x}_i; 0]$ and $\overline{\mathbf{h}}_i^y = \mathbf{W}[\mathbf{0}_D; y_i]$; (2) *Linear-copy* embedding, where the y tokens are instead $\overline{\mathbf{h}}_i^y = \mathbf{W}[\mathbf{x}_i; y_i]$, motivated by the format (4); (3) *One-layer transformer* embedding $\overline{\mathrm{TF}}$, which computes $\overline{\mathbf{H}} = \overline{\mathrm{TF}}(\mathbf{H})$. See Appendix D.2 for further setups of pasting.

Figure 4b shows the pasting results on a trained transformer on $(L, D, \sigma) = (3, 20, 0.1)$ (an ablation in Figure 10b), where we dissect the lower and upper modules at layer 4 as suggested by the probing curve (Figure 3a green). Perhaps surprisingly, the upper module of the transformer can indeed perform nearly optimal linear ICL without representation when we use the one-layer transformer

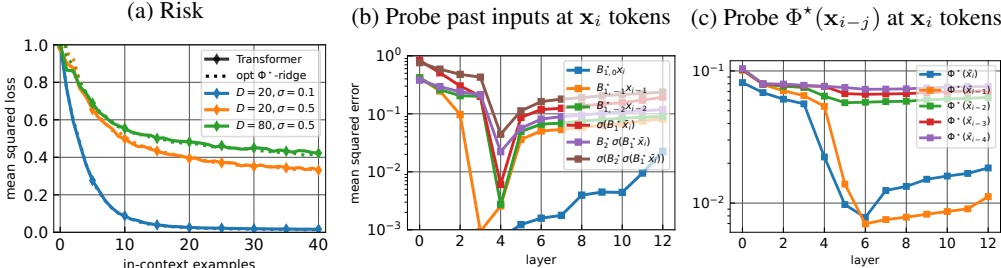

Figure 5: ICL risks and probing errors for the dynamical system setting. **(a)** Each curve modifies problem parameters from the base setting $(k, L, D, \sigma) = (3, 2, 80, 0.5)$. **(b,c)** Results are with the same base setting.

embedding. Note that a (freshly trained) single-layer transformer itself performs badly, achieving about the trivial test risk 1.01, which is expected due to our specific input format[3] (1). This suggests that the majority of the ICL is indeed carried by the upper module, with the one-layer transformer embedding not doing much ICL itself. Also note that the linear-copy and linear embeddings also yield reasonable (though suboptimal) performance, with linear-copy performing slightly better.

### 5.1.1 EXTENSION: MIXTURE OF MULTIPLE REPRESENTATIONS

We aditionally investigate an harder scenario in which there exists *multiple possible representation functions* $(\Phi_j^\star)_{j \in [K]}$, and the ICL data distribution is a mixture of the $K$ distributions of form (2) each induced by $\Phi_j^\star$ (equivalent to using the concatenated representation $\overline{\Phi}^\star = [\Phi_1^\star, \ldots, \Phi_K^\star]$ with a group 1-sparse prior on $\overline{\mathbf{w}} \in \mathbb{R}^{KD}$). We find that transformers still approach Bayes-optimal risks, though less so compared with the single-representation setting. Using linear probes, we find that transformers sometimes implement the *post-ICL algorithm selection* mechanism identified in Bai et al. (2023), depending on the setting. Details are deferred to Appendix E due to the space limit.

### 5.2 DYNAMICAL SYSTEMS

We now study the dynamical systems setting in Section 4.2 using the same approaches as in Section 5.1. Figure 5a shows that transformers can still consistently achieve nearly Bayes-optimal ICL risk. An ablation of the risks and probing errors in alternative settings can be found in Appendix F.2.

**Probing copying mechanisms**  The main mechanistic question we ask here is about the data preparation phase, where the transformer construction in Theorem 2 performs copying *twice*:

  i) A copying of $[\mathbf{x}_{i-k+1}; \ldots; \mathbf{x}_{i-1}]$ onto the $\mathbf{x}_i$ token as in (7), to prepare for the computation of $\Phi^\star(\overline{\mathbf{x}}_i)$; As copying may not be distinguishable from the consequent *matrix multiplication* step $[\mathbf{x}_{i-k+1}; \ldots; \mathbf{x}_{i-1}; \mathbf{x}_i] \mapsto \mathbf{B}_1^\star[\mathbf{x}_{i-k+1}; \ldots; \mathbf{x}_{i-1}; \mathbf{x}_i]$, we probe instead the result $\mathbf{B}_{1,-j}^\star \mathbf{x}_{i-j}$ after matrix multiplication, where $\mathbf{B}_{1,-j}^\star \in \mathbb{R}^{D \times d}$ denotes the block within $\mathbf{B}_1^\star$ hitting $\mathbf{x}_{i-j}$.

 ii) A second copying of $\Phi^\star(\overline{\mathbf{x}}_{i-1})$ onto the $\mathbf{x}_i$ token to obtain (9), after $\{\Phi^\star(\overline{\mathbf{x}}_i)\}_i$ are computed.

We probe one transformer trained on the dynamical systems problem with $k = 3$ (so that the useful preceding inputs are $\mathbf{x}_{i-1}$ and $\mathbf{x}_{i-2}$), and find that the transformer indeed performs the two conjectured copyings. Figure 5b demonstrates copying i) onto the current token, where the copying of $\mathbf{x}_{i-1}$ happens earlier (at layer 3) and is slightly more accurate than that of $\mathbf{x}_{i-2}$ (at layer 4), as expected. Further observe that layer 4 (which we recall contains an attention layer and an MLP layer) have seemingly also implemented the (unnormalized) MLP representation $\widetilde{\Phi}^\star(\overline{\mathbf{x}}_i) = \sigma_\rho(\mathbf{B}_2^\star \sigma_\rho(\mathbf{B}_1^\star \overline{\mathbf{x}}_i))$, though the probing error for the actual representation $\Phi^\star(\overline{\mathbf{x}}_i) = \widetilde{\Phi}^\star(\overline{\mathbf{x}}_i) / \|\widetilde{\Phi}^\star(\overline{\mathbf{x}}_i)\|_2$ continues to drop in layer 4-6 (Figure 5c). Figure 5c further demonstrates copying ii), where $\Phi^\star(\overline{\mathbf{x}}_{i-1})$ are indeed copied to the $i$-th token, whereas by sharp contrast $\Phi^\star(\overline{\mathbf{x}}_{i-k})$ for $k \geq 2$ are *not* copied at all into the $\mathbf{x}_i$ token, aligning with our conjectured intermediate output format (9).

---

[3]A one-layer transformer does not have much ICL power using input format (1)—$\mathbf{x}_i$ and $y_i$ are stored in separate tokens there, which makes "one-layer" mechanisms such as gradient descent (von Oswald et al., 2022; Akyürek et al., 2022; Bai et al., 2023) unlikely to be implementable; see Appendix D.3 for a discussion.

## 6 CONCLUSION

This paper presents theoretical and mechanistic studies on the in-context learning ability of transformers on learning tasks involving representation functions, where we give efficient transformer constructions for linear ICL on top of representations for the supervised learning and dynamical system setting, and empirically confirm the existence of various high-level mechanisms in trained transformers. We believe our work opens up the investigation of ICL beyond simple function classes, and suggests open questions such as further investigations of the mechanisms of the linear ICL modules, and theory for ICL in more complex function classes. One limitation of our work is that the setting still consists of synthetic data with idealistic representation functions; performing similar studies on more real-world data would be an important direction for future work.

## ACKNOWLEDGMENT

WH acknowledges support from the Google Research Scholar program. S. Mei is supported by NSF DMS-2210827, CCF-2315725, NSF CAREER DMS-2339904, and an Amazon Research Award.

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

## A  TECHNICAL TOOLS

The following convergence result for minimizing a smooth and strongly convex function is standard from the convex optimization literature, e.g. by adapting the learning rate in Bubeck (2015, Theorem 3.10) from $\eta = 1/\beta$ to any $\eta \le 1/\beta$.

**Proposition A.1** (Gradient descent for smooth and strongly convex functions)**.** *Suppose $L : \mathbb{R}^d \to \mathbb{R}$ is $\alpha$-strongly convex and $\beta$-smooth for some $0 < \alpha \le \beta$. Then, the gradient descent iterates $\mathbf{w}_{\mathrm{GD}}^{t+1} := \mathbf{w}_{\mathrm{GD}}^t - \eta\nabla L(\mathbf{w}_{\mathrm{GD}}^t)$ with learning rate $\eta \le 1/\beta$ and initialization $\mathbf{w}_{\mathrm{GD}}^0 \in \mathbb{R}^d$ satisfies for any $t \ge 1$,*

$$\left\|\mathbf{w}_{\mathrm{GD}}^t - \mathbf{w}^\star\right\|_2^2 \le \exp\left(-\eta\alpha \cdot t\right) \cdot \left\|\mathbf{w}_{\mathrm{GD}}^0 - \mathbf{w}^\star\right\|_2^2.$$

*where $\mathbf{w}^\star := \arg\min_{\mathbf{w}\in\mathbb{R}^d} L(\mathbf{w})$ is the minimizer of $L$.*

## B  PROOFS FOR SECTION 4.1

Throughout the rest of this and next section, we consider transformer architectures defined in Section 3 where we choose $\overline{\sigma}$ to be the (entry-wise) ReLU activation normalized by sequence length, following (Bai et al., 2023): For all $\mathbf{A} \in \mathbb{R}^{N \times N}$ and $i, j \in [N]$,

$$[\overline{\sigma}(\mathbf{A})]_{ij} = \frac{1}{j}\sigma(A_{ij}), \tag{11}$$

where we recall $\sigma(t) = \max\{t, 0\}$ denotes the standard ReLU. This activation is similar as the softmax in that, for every (query index) $j$, the resulting attention weights $\left\{\frac{1}{j}\sigma(A_{ij})\right\}_{i\in[j]}$ is approximately a probability distribution in typical scenarios, in the sense that they are non-negative and sum to $O(1)$ when each $A_{ij} = O(1)$. We remark that transformers with (normalized) ReLU activation is

recently shown to achieve comparable performance with softmax in larger-scale tasks (Shen et al., 2023; Wortsman et al., 2023).

With activation chosen as (11), a (decoder-only) attention layer $\widetilde{\mathbf{H}} = \text{Attn}_{\boldsymbol{\theta}}(\mathbf{H})$ with $\boldsymbol{\theta} = (\mathbf{Q}_m, \mathbf{K}_m, \mathbf{V}_m)_{m \in [M]}$ takes the following form in vector notation:

$$\widetilde{\mathbf{h}}_i = \mathbf{h}_i + \sum_{m=1}^{M} \frac{1}{i} \sum_{j=1}^{i} \sigma(\langle \mathbf{Q}_m \mathbf{h}_i, \mathbf{K}_m \mathbf{h}_j \rangle) \cdot \mathbf{V}_m \mathbf{h}_j.$$

Recall our input format (1):

$$\mathbf{H} = \begin{bmatrix} \mathbf{x}_1 & \mathbf{0} & \dots & \mathbf{x}_N & \mathbf{0} \\ 0 & y_1 & \dots & 0 & y_N \\ \mathbf{p}_1^x & \mathbf{p}_1^y & \dots & \mathbf{p}_N^x & \mathbf{p}_N^y \end{bmatrix} \in \mathbb{R}^{D_{\text{hid}} \times 2N}.$$

We will use $(\mathbf{h}_k)_{k \in [2N]}$ and $(\mathbf{h}_i^x, \mathbf{h}_i^y)_{i \in [N]}$ interchangeably to denote the tokens in (1), where $\mathbf{h}_i^x := \mathbf{h}_{2i-1}$ and $\mathbf{h}_i^y := \mathbf{h}_{2i}$. Similarly, we will use $(\mathbf{p}_i^x, \mathbf{p}_i^y)_{i \in [N]}$ and $(\mathbf{p}_k)_{k \in [2N]}$ interchangably to denote the positional encoding vectors in (1), where $\mathbf{p}_{2i-1} := \mathbf{p}_i^x$ and $\mathbf{p}_{2i} := \mathbf{p}_i^y$. Unless otherwise specified, we typically reserve use $i, j$ as (query, key) indices within $[N]$ and $k, \ell$ as (query, key) indices within $[2N]$.

We use the following positional encoding vectors for all $i \in [N]$:

$$\begin{aligned} \mathbf{p}_i^x &= [\mathbf{0}_{D_{\text{hid}}-d-9}; 1; 2i-1; (2i-1)^2; (2i-1)^3; i; i^2; 1; i], \\ \mathbf{p}_i^y &= [\mathbf{0}_{D_{\text{hid}}-d-9}; 1; 2i; (2i)^2; (2i)^3; i; i^2; 0; 0]. \end{aligned} \tag{12}$$

Note that $\mathbf{p}_k$ contains $[1; k; k^2; k^3]$ for all $k \in [2N]$; $\mathbf{p}_i^x, \mathbf{p}_i^y$ contains $[i; i^2]$, an indicator of being an $x$-token, and the product of the indicator and $i$.

## B.1 USEFUL TRANSFORMER CONSTRUCTIONS

**Lemma B.1** (Copying by a single attention head). *There exists a single-head attention layer $\boldsymbol{\theta} = (\mathbf{Q}, \mathbf{K}, \mathbf{V}) \subset \mathbb{R}^{D_{\text{hid}} \times D_{\text{hid}}}$ that copies each $\mathbf{x}_i$ into the next token for every input $\mathbf{H}$ of the form (1), i.e.*

$$\text{Attn}_{\boldsymbol{\theta}}(\mathbf{H}) = \begin{bmatrix} \mathbf{x}_1 & \mathbf{x}_1 & \dots & \mathbf{x}_N & \mathbf{x}_N \\ 0 & y_1 & \dots & 0 & y_N \\ \mathbf{p}_1^x & \mathbf{p}_1^y & \dots & \mathbf{p}_N^x & \mathbf{p}_N^y \end{bmatrix} \in \mathbb{R}^{D_{\text{hid}} \times 2N}.$$

*Proof.* By assumption of the positional encoding vectors, we can define matrices $\mathbf{Q}, \mathbf{K} \in \mathbb{R}^{D_{\text{hid}} \times D_{\text{hid}}}$ such that for all $k, \ell \in [2N]$,

$$\mathbf{Q} \mathbf{h}_k = [k^3; k^2; k; \mathbf{0}_{D_{\text{hid}}-3}], \quad \mathbf{K} \mathbf{h}_\ell = [-1; 2\ell+2; -\ell^2 - 2\ell; \mathbf{0}_{D_{\text{hid}}-3}].$$

This gives that for all $\ell \le k$,

$$\begin{aligned} &\sigma(\langle \mathbf{Q} \mathbf{h}_k, \mathbf{K} \mathbf{h}_\ell \rangle) \\ &= \sigma\big(-k^3 + k^2(2\ell+2) - k(\ell^2 + 2\ell)\big) = \sigma\big(k(1 - (k - \ell - 1)^2)\big) = k \mathbb{1}\{\ell = k-1\}. \end{aligned}$$

Further defining $\mathbf{V}$ such that $\mathbf{V} \mathbf{h}_i^x = [\mathbf{x}_i; \mathbf{0}]$ and $\mathbf{V} \mathbf{h}_i^y = \mathbf{0}$, we have for every $k \in [2N]$ that

$$\begin{aligned} &\sum_{\ell \le k} \frac{1}{k} \sigma(\langle \mathbf{Q} \mathbf{h}_k, \mathbf{K} \mathbf{h}_\ell \rangle) \mathbf{V} \mathbf{h}_\ell \\ &= \frac{1}{k} \cdot k \mathbb{1}\{\ell = k-1\} \cdot [\mathbf{x}_{\lceil \ell/2 \rceil} \mathbb{1}\{\ell \text{ is odd}\}; \mathbf{0}] = [\mathbf{x}_{\lceil \ell/2 \rceil}; \mathbf{0}] \cdot \mathbb{1}\{\ell = k-1 \text{ and } \ell \text{ is odd}\}. \end{aligned}$$

By the residual structure of the attention layer, the above exactly gives the desired copying behavior, where every $\mathbf{x}_i$ on the odd token $\mathbf{H}$ is copied to the next token. $\square$

**Lemma B.2** (Linear prediction layer). *For any $B_x, B_w, B_y > 0$, there exists an attention layer $\boldsymbol{\theta} = \{(\mathbf{Q}_m, \mathbf{K}_m, \mathbf{V}_m)\}_{m \in [M]}$ with $M = 2$ heads such that the following holds. For any input sequence $\mathbf{H} \in \mathbb{R}^{D_{\mathrm{hid}} \times 2N}$ that takes form*

$$\mathbf{h}_i^x = [\mathbf{x}_i; 0; \mathbf{w}_i; \mathbf{p}_i^x], \quad \mathbf{h}_i^y = [\mathbf{x}_i; y_i; \mathbf{0}_d; \mathbf{p}_i^y]$$

*with $\|\mathbf{x}_i\|_2 \le B_x$, $|y_i| \le B_y$, and $\|\mathbf{w}\|_2 \le B_w$, it gives output $\mathrm{Attn}_{\boldsymbol{\theta}}(\mathbf{H}) = \widetilde{\mathbf{H}} \in \mathbb{R}^{D_{\mathrm{hid}} \times 2N}$ with*

$$\widetilde{\mathbf{h}}_i^x = \widetilde{\mathbf{h}}_{2i-1} = [\mathbf{x}_i; \widehat{y}_i; \mathbf{w}_i; \mathbf{p}_i^x], \quad \text{where } \widehat{y}_i = \langle \mathbf{x}_i, \mathbf{w}_i \rangle$$

*for all $i \in [N]$.*

*Proof.* Let $R := \max \{B_x B_w, B_y\}$. Define matrices $(\mathbf{Q}_m, \mathbf{K}_m, \mathbf{V}_m)_{m=1,2}$ as

$$\mathbf{Q}_1 \mathbf{h}_i^x = \begin{bmatrix} \mathbf{w}_i \\ i \\ R \\ \mathbf{0} \end{bmatrix}, \quad \mathbf{K}_1 \mathbf{h}_j^x = \mathbf{K}_1 \mathbf{h}_j^y = \begin{bmatrix} \mathbf{x}_j \\ -2R \\ 2j+1 \\ \mathbf{0} \end{bmatrix}, \mathbf{V}_1 \mathbf{h}_\ell = \begin{bmatrix} \mathbf{0}_d \\ \ell \\ \mathbf{0}_{D_{\mathrm{hid}}-d-1} \end{bmatrix},$$

$$\mathbf{Q}_2 \mathbf{h}_i^x = \begin{bmatrix} i \\ R \\ \mathbf{0} \end{bmatrix}, \quad \mathbf{K}_2 \mathbf{h}_j^x = \mathbf{K}_1 \mathbf{h}_j^y = \begin{bmatrix} -2R \\ 2j+1 \\ \mathbf{0} \end{bmatrix}, \mathbf{V}_2 \mathbf{h}_\ell = -\begin{bmatrix} \mathbf{0}_d \\ \ell \\ \mathbf{0}_{D_{\mathrm{hid}}-d-1} \end{bmatrix}$$

for all $i, j \in [N]$ and $\ell \in [2N]$. For every $i \in [N]$, we then have

$$\sum_{m=1}^{2} \sum_{\ell=1}^{2i-1} \frac{1}{2i-1} \sigma(\langle \mathbf{Q}_m \mathbf{h}_i^x, \mathbf{K}_m \mathbf{h}_\ell \rangle) \cdot \mathbf{V}_m \mathbf{h}_\ell$$

$$= \frac{1}{2i-1} \Bigg( \sum_{j=1}^{i} \left[ \sigma\big(\mathbf{w}_i^\top \mathbf{x}_j + R(-2i+2j+1)\big) - \sigma(R(-2i+2j+1)) \right] \cdot [\mathbf{0}_d; 2j-1; \mathbf{0}_{D_{\mathrm{hid}}-d-1}]$$

$$+ \sum_{j=1}^{i-1} \left[ \sigma\big(\mathbf{w}_i^\top \mathbf{x}_j + R(-2i+2j-1)\big) - \sigma(R(-2i+2j+1)) \right] \cdot [\mathbf{0}_d; 2j; \mathbf{0}_{D_{\mathrm{hid}}-d-1}] \Bigg)$$

$$= \frac{1}{2i-1} \cdot \mathbf{w}_i^\top \mathbf{x}_i \cdot [\mathbf{0}_d; 2i-1; \mathbf{0}_{D_{\mathrm{hid}}-d-1}] = [\mathbf{0}_d; \mathbf{w}_i^\top \mathbf{x}_i; \mathbf{0}_{D_{\mathrm{hid}}-d-1}].$$

By the residual structure of an attention layer, the above shows the desired result. $\qquad \square$

**Lemma B.3** (Implementing MLP representation by transformers). *Fix any MLP representation function $\Phi^\star$ of the form (3), suppose $D_{\mathrm{hid}} \ge \max \{2D, D + d + 10\}$, where $D$ is the hidden dimension within the MLP (3). Then there exists a transformer $\mathrm{TF}_{\boldsymbol{\theta}}$ with $(L+1)$ layers and 5 heads that exactly implements $\Phi^\star$ in a token-wise fashion, i.e. for any input $\mathbf{H}$ of form (1),*

$$\widetilde{\mathbf{H}} = \mathrm{TF}_{\boldsymbol{\theta}}(\mathbf{H}) = \begin{bmatrix} \Phi^\star(\mathbf{x}_1) & \mathbf{0} & \dots & \Phi^\star(\mathbf{x}_N) & \mathbf{0} \\ 0 & y_1 & \dots & 0 & y_N \\ \widetilde{\mathbf{p}}_1^x & \widetilde{\mathbf{p}}_1^y & \dots & \widetilde{\mathbf{p}}_N^x & \widetilde{\mathbf{p}}_N^y \end{bmatrix},$$

*where $\widetilde{\mathbf{p}}_i^x, \widetilde{\mathbf{p}}_i^y$ differs from $\mathbf{p}_i^x, \mathbf{p}_i^y$ only in the dimension of their zero paddings.*

*Proof.* Recall that $\Phi^\star(\mathbf{x}) = \sigma_\rho(\mathbf{B}_L^\star \cdots \sigma_\rho(\mathbf{B}_1^\star \mathbf{x}) \cdots)$. We first show how to implement a single MLP layer $\mathbf{x} \mapsto \sigma_\rho(\mathbf{B}_1^\star \mathbf{x})$ by an (MLP-Attention) structure.

Consider any input token $\mathbf{h}_i^x = [\mathbf{x}_i; 0; \mathbf{p}_i^x]$ at an $x$-location. Define matrices $\mathbf{W}_1, \mathbf{W}_2 \in \mathbb{R}^{D_{\mathrm{hid}} \times D_{\mathrm{hid}}}$ such that

$$\mathbf{W}_1 \mathbf{h}_i^x = \begin{bmatrix} \mathbf{B}_1^\star \mathbf{x}_i \\ -\mathbf{B}_1^\star \mathbf{x}_i \\ \mathbf{0} \end{bmatrix}, \quad \sigma(\mathbf{W}_1 \mathbf{h}_i^x) = \begin{bmatrix} \sigma(\mathbf{B}_1^\star \mathbf{x}_i) \\ \sigma(-\mathbf{B}_1^\star \mathbf{x}_i) \\ \mathbf{0} \end{bmatrix},$$

$$\mathbf{W}_2 \sigma(\mathbf{W}_1 \mathbf{h}_i^x) = \begin{bmatrix} \mathbf{0}_d \\ \sigma(\mathbf{B}_1^\star \mathbf{x}_i) - \rho \sigma(-\mathbf{B}_1^\star \mathbf{x}_i) \\ \mathbf{0} \end{bmatrix} = \begin{bmatrix} \mathbf{0}_d \\ \sigma_\rho(\mathbf{B}_1^\star \mathbf{x}_i) \\ \mathbf{0} \end{bmatrix}.$$

Therefore, the MLP layer $(\mathbf{W}_1, \mathbf{W}_2)$ outputs

$$\overline{\mathbf{h}}_i^x := [\text{MLP}_{\mathbf{W}_1, \mathbf{W}_2}(\mathbf{H})]_i^x = \mathbf{h}_i^x + \mathbf{W}_2 \sigma(\mathbf{W}_1 \mathbf{h}_i^x) = \begin{bmatrix} \mathbf{x}_i \\ \sigma_\rho(\mathbf{B}_1^\star \mathbf{x}_i) \\ \mathbf{0} \\ \mathbf{p}_i^x \end{bmatrix},$$

and does not change the $y$-tokens.

We next define an attention layer that "moves" $\sigma_\rho(\mathbf{B}_1 \mathbf{x}_i)$ to the beginning of the token, and removes $\mathbf{x}_i$. Define three attention heads $\boldsymbol{\theta} = (\mathbf{Q}_m, \mathbf{K}_m, \mathbf{V}_m)_{m \in [3]}$ as follows:

$$\mathbf{Q}_{\{1,2,3\}} \overline{\mathbf{h}}_k = \begin{bmatrix} k^2 \\ k \\ k\mathbf{1}\{k \text{ is odd}\} \\ \mathbf{0} \end{bmatrix}, \mathbf{K}_{\{1,2,3\}} \overline{\mathbf{h}}_\ell = \begin{bmatrix} -1 \\ \ell \\ 1 \\ \mathbf{0} \end{bmatrix},$$

$$\mathbf{V}_1 \overline{\mathbf{h}}_j^x = \begin{bmatrix} \sigma_\rho(\mathbf{B}_1^\star \mathbf{x}_j) \\ \mathbf{0}_d \\ \mathbf{0} \end{bmatrix}, \mathbf{V}_2 \overline{\mathbf{h}}_j^x = \begin{bmatrix} -\mathbf{x}_j \\ \mathbf{0}_D \\ \mathbf{0} \end{bmatrix}, \mathbf{V}_3 \overline{\mathbf{h}}_j^x = \begin{bmatrix} \mathbf{0}_d \\ -\sigma_\rho(\mathbf{B}_1^\star \mathbf{x}_j) \\ \mathbf{0} \end{bmatrix}.$$

The values for $\mathbf{V}_{1,2,3} \overline{\mathbf{h}}_i^y$ are defined automatically by the same operations over the $\overline{\mathbf{h}}_i^y$ tokens (which does not matter to the proof, as we see shortly). For any $\ell \le k$ and $m \in [3]$,

$$\frac{1}{k} \sigma(\langle \mathbf{Q}_m \overline{\mathbf{h}}_k, \mathbf{K}_m \overline{\mathbf{h}}_\ell \rangle) = \frac{1}{k} \sigma(k(-k + \ell + \mathbf{1}\{k \text{ is odd}\})) = \mathbf{1}\{\ell = k, \ k \text{ is odd}\}.$$

Therefore, these three attention heads are only active iff the query token $k = 2i - 1$ is odd (i.e. being an $x$-token) and $\ell = k = 2i - 1$. At such tokens, the three value matrices (combined with the residual structure of attention) would further remove the $\mathbf{x}_i$ part, and move $\sigma_\rho(\mathbf{B}_1^\star \mathbf{x}_i)$ to the beginning of the token, i.e.

$$\widetilde{\mathbf{h}}_i^x = [\text{Attn}_{\boldsymbol{\theta}}(\overline{\mathbf{H}})]_i^x = \begin{bmatrix} \sigma_\rho(\mathbf{B}_1^\star \mathbf{x}_i) \\ 0 \\ \mathbf{p}_i^x \end{bmatrix},$$

and $\widetilde{\mathbf{h}}_i^y = \mathbf{h}_i^y$. Additionally, we now add two more attention heads into $\boldsymbol{\theta}$ to move all $y_i$ from entry $d + 1$ to $D + 1$, and leaves the $x$-tokens unchanged.

Repeating the above argument $L$ times, we obtain a structure (MLP-Attention-...-MLP-Attention) with five heads in each attention layer that exactly implements the $\Phi^\star$ in a token-wise fashion. This structure can be rewritten as an $(L+1)$-layer transformer by appending an identity {Attention, MLP} layer (with zero weights) {before, after} the structure respectively, which completes the proof. $\square$

## B.2 IN-CONTEXT RIDGE REGRESSION BY DECODER TRANSFORMER

This section proves the existence of a decoder transformer that approximately implements in-context ridge regression at every token $i \in [N]$ simultaneously. For simplicity, we specialize our results to the ridge regression problem; however, our construction can be directly generalized to any (generalized) linear models with a sufficiently smooth loss, by approximating the gradient of the loss by sum of relus (Bai et al., 2023, Section 3.5).

Denote the regularized empirical risk for ridge regression on dataset $\mathcal{D}_i = \{(\mathbf{x}_j, y_j)\}_{j \in [i]}$ by

$$\widehat{L}_i^\lambda(\mathbf{w}) := \frac{1}{2i} \sum_{j=1}^i (\mathbf{w}^\top \mathbf{x}_j - y_j)^2 + \frac{\lambda}{2} \|\mathbf{w}\|_2^2 \tag{13}$$

for all $i \in [N]$. Let $\widehat{\mathbf{w}}_i^\lambda := \arg\min_{\mathbf{w} \in \mathbb{R}^d} \widehat{L}_{i-1}^\lambda(\mathbf{w})$ denote the minimizer of the above risk (solution of ridge regression) for dataset $\mathcal{D}_{i-1}$. We further understand $\widehat{L}_0^\lambda(\mathbf{w}) := 0$ and $\widehat{\mathbf{w}}_1^\lambda := \mathbf{0}$. Let $\widehat{L}_i(\mathbf{w}) := \widehat{L}_i^0(\mathbf{w})$ denote the unregularized version of the above risk.

**Proposition B.4** (Approximating a single GD step by a single attention layer)**.** *For any $\eta > 0$ and any $B_x, B_w, B_y > 0$, there exists an attention layer $\boldsymbol{\theta} = \{(\mathbf{Q}_m, \mathbf{K}_m, \mathbf{V}_m)\}_{m \in [M]}$ with $M = 3$ heads such that the following holds. For any input sequence $\mathbf{H} \in \mathbb{R}^{D_{\text{hid}} \times 2N}$ that takes form*

$$\mathbf{h}_i^x = [\mathbf{x}_i; 0; \mathbf{w}_i; \mathbf{p}_i^x], \quad \mathbf{h}_i^y = [\mathbf{x}_i; y_i; \mathbf{0}_d; \mathbf{p}_i^y]$$

*with* $\|\mathbf{x}_i\|_2 \leq B_x$, $|y_i| \leq B_y$, *and* $\|\mathbf{w}\|_2 \leq B_w$, *it gives output* $\mathrm{Attn}_{\boldsymbol{\theta}}(\mathbf{H}) = \widetilde{\mathbf{H}} \in \mathbb{R}^{D_{\mathrm{hid}} \times 2N}$ *with* $\widetilde{\mathbf{h}}_i^x = \widetilde{\mathbf{h}}_{2i-1} = [\mathbf{x}_i; 0; \widetilde{\mathbf{w}}_i; \mathbf{p}_i^x]$, *where*

$$\widetilde{\mathbf{w}}_i = \mathbf{w}_i - \eta_i \nabla \widehat{L}_{i-1}^{\lambda}(\mathbf{w}_i)$$

*with* $\eta_i = \frac{i-1}{2i-1}\eta$, *and* $\widetilde{\mathbf{h}}_i^y = \mathbf{h}_i^y$, *for all* $i \in [N]$.

*Proof.* Let $R := \max\{B_x B_w, B_y\}$. By the form of the input $(\mathbf{h}_k)_{k \in [2N]}$ in (1), we can define two attention heads $\{(\mathbf{Q}_m, \mathbf{K}_m, \mathbf{V}_m)\}_{m=1,2} \subset \mathbb{R}^{D_{\mathrm{hid}} \times D_{\mathrm{hid}}}$ such that for all $i, j \in [N]$,

$$\mathbf{Q}_1 \mathbf{h}_i^x = \begin{bmatrix} \mathbf{w}_i/2 \\ -1 \\ i \\ -3R \\ -R \\ \mathbf{0} \end{bmatrix}, \quad \mathbf{K}_1 \mathbf{h}_j^y = \begin{bmatrix} \mathbf{x}_j \\ y_j \\ 3R \\ j \\ 1 \\ \mathbf{0} \end{bmatrix}, \quad \mathbf{V}_1 \mathbf{h}_j^x = \mathbf{V}_1 \mathbf{h}_j^y = -\eta \cdot \begin{bmatrix} \mathbf{0}_{d+1} \\ \mathbf{x}_j \\ \mathbf{0}_{D_{\mathrm{hid}}-2d-1} \end{bmatrix},$$

$$\mathbf{Q}_2 \mathbf{h}_i^x = \mathbf{Q}_2 \mathbf{h}_i^y = \begin{bmatrix} i \\ -3R \\ -R \\ \mathbf{0} \end{bmatrix}, \quad \mathbf{K}_2 \mathbf{h}_j^x = \mathbf{K}_2 \mathbf{h}_j^y = \begin{bmatrix} 3R \\ j \\ 1 \\ \mathbf{0} \end{bmatrix}, \quad \mathbf{V}_2 \mathbf{h}_j^x = \mathbf{V}_2 \mathbf{h}_j^y = \eta \cdot \begin{bmatrix} \mathbf{0}_{d+1} \\ \mathbf{x}_j \\ \mathbf{0}_{D_{\mathrm{hid}}-2d-1} \end{bmatrix}.$$

Further, $\mathbf{Q}_1 \mathbf{h}_i^y$ takes the same form as $\mathbf{Q}_1 \mathbf{h}_i^x$ except for replacing the $\mathbf{w}_i/2$ location with $\mathbf{0}_d$ and replacing the $-1$ location with $0$ (using the indicator for being an $x$-token within $\mathbf{p}_i^x, \mathbf{p}_i^y$); $\mathbf{K}_1 \mathbf{h}_j^x$ takes the same form as $\mathbf{K}_1 \mathbf{h}_j^y$ except for replacing the $y_j$ location with $0$.

Fixing any $i \in [N]$. We have for all $j \leq i - 1$,

$$\sigma\big(\langle \mathbf{Q}_1 \mathbf{h}_i^x, \mathbf{K}_1 \mathbf{h}_j^y \rangle\big) - \sigma\big(\langle \mathbf{Q}_2 \mathbf{h}_i^x, \mathbf{K}_2 \mathbf{h}_j^y \rangle\big)$$
$$= \sigma\big(\mathbf{w}_i^\top \mathbf{x}_j/2 - y_j + R(3i - 3j - 1)\big) - \sigma\big(R(3i - 3j - 1)\big) = \mathbf{w}_i^\top \mathbf{x}_j/2 - y_j,$$

and for all $j \leq i$,

$$\sigma\big(\langle \mathbf{Q}_1 \mathbf{h}_i^x, \mathbf{K}_1 \mathbf{h}_j^x \rangle\big) - \sigma\big(\langle \mathbf{Q}_2 \mathbf{h}_i^x, \mathbf{K}_2 \mathbf{h}_j^x \rangle\big)$$
$$= \sigma\big(\mathbf{w}_i^\top \mathbf{x}_j/2 + R(3i - 3j - 1)\big) - \sigma\big(R(3i - 3j - 1)\big) = \mathbf{w}_i^\top \mathbf{x}_j/2 \cdot \mathbb{1}\{j \leq i - 1\}.$$

Above, we have used $|\mathbf{w}_i^\top \mathbf{x}_j/2 - y_j| \leq 3R/2$, $|\mathbf{w}_i^\top \mathbf{x}_j/2| \leq R/2$, and the fact that $\sigma(z + M) - \sigma(M)$ equals $z$ for $M \geq |z|$ and $0$ for $M \leq -|z|$.

Therefore for all $j \leq i - 1$,

$$\sigma\big(\langle \mathbf{Q}_1 \mathbf{h}_i^x, \mathbf{K}_1 \mathbf{h}_j^y \rangle\big) \mathbf{V}_1 \mathbf{h}_j^y + \sigma\big(\langle \mathbf{Q}_2 \mathbf{h}_i^x, \mathbf{K}_2 \mathbf{h}_j^y \rangle\big) \mathbf{V}_2 \mathbf{h}_j^y$$
$$= \big(\sigma\big(\langle \mathbf{Q}_1 \mathbf{h}_i^x, \mathbf{K}_1 \mathbf{h}_j^y \rangle\big) - \sigma\big(\langle \mathbf{Q}_2 \mathbf{h}_i^x, \mathbf{K}_2 \mathbf{h}_j^y \rangle\big)\big) \cdot -\eta[\mathbf{0}_{d+1}; \mathbf{x}_j; \mathbf{0}_{D_{\mathrm{hid}}-2d-1}]$$
$$= -\eta\big(\mathbf{w}_i^\top \mathbf{x}_j/2 - y_j\big) \cdot [\mathbf{0}_{d+1}; \mathbf{x}_j; \mathbf{0}_{D_{\mathrm{hid}}-2d-1}],$$

and similarly for all $j \leq i$,

$$\sigma\big(\langle \mathbf{Q}_1 \mathbf{h}_i^x, \mathbf{K}_1 \mathbf{h}_j^x \rangle\big) \mathbf{V}_1 \mathbf{h}_j^x + \sigma\big(\langle \mathbf{Q}_2 \mathbf{h}_i^x, \mathbf{K}_2 \mathbf{h}_j^x \rangle\big) \mathbf{V}_2 \mathbf{h}_j^x$$
$$= -\eta\big(\mathbf{w}_i^\top \mathbf{x}_j/2\big) \mathbb{1}\{j \leq i - 1\} \cdot [\mathbf{0}_{d+1}; \mathbf{x}_j; \mathbf{0}_{D_{\mathrm{hid}}-2d-1}]$$

Summing the above over all key tokens $\ell \in [2i - 1]$, we obtain the combined output of the two heads at query token $2i - 1$ (i.e. the $i$-th $x$-token):

$$\sum_{\ell=1}^{2i-1} \sum_{m=1,2} \frac{1}{2i-1} \sigma(\langle \mathbf{Q}_m \mathbf{h}_{2i-1}, \mathbf{K}_m \mathbf{h}_\ell \rangle) \mathbf{V}_m \mathbf{h}_\ell$$
$$= \sum_{j=1}^{i-1} \sum_{m=1,2} \frac{1}{2i-1} \sigma(\langle \mathbf{Q}_m \mathbf{h}_i^x, \mathbf{K}_m \mathbf{h}_j^y \rangle) \mathbf{V}_m \mathbf{h}_j^y + \sum_{j=1}^{i} \sum_{m=1,2} \frac{1}{2i-1} \sigma(\langle \mathbf{Q}_m \mathbf{h}_i^x, \mathbf{K}_m \mathbf{h}_j^x \rangle) \mathbf{V}_m \mathbf{h}_j^x$$
$$= \frac{1}{2i-1} \left[\sum_{j=1}^{i-1} -\eta\big(\mathbf{w}_i^\top \mathbf{x}_j/2 - y_j\big) + \sum_{j=1}^{i} -\eta\big(\mathbf{w}_i^\top \mathbf{x}_j/2\big) \mathbb{1}\{j \leq i - 1\}\right] \cdot [\mathbf{0}_{d+1}; \mathbf{x}_j; \mathbf{0}_{D_{\mathrm{hid}}-2d-1}]$$
$$= \frac{i-1}{2i-1} \cdot \left[\mathbf{0}_{d+1}; -\eta \nabla \widehat{L}_{i-1}(\mathbf{w}_i); \mathbf{0}_{D_{\mathrm{hid}}-2d-1}\right].$$

$$\tag{14}$$

It is straightforward to see that, repeating the same operation at query token $2i$ (i.e. the $i$-th $y$-token) would output $\mathbf{0}_{D_{\mathrm{hid}}}$, since the query vector $\mathbf{Q}_1 \mathbf{h}_i^y$ contains $[\mathbf{0}_d; 0]$ instead of $[\mathbf{w}_i/2; -1]$ as in $\mathbf{Q}_1 \mathbf{h}_i^x$.

We now define one more attention head $(\mathbf{Q}_3, \mathbf{K}_3, \mathbf{V}_3) \subset \mathbb{R}^{D \times D}$ such that for all $k \in [2N]$, $j \in [N]$,

$$\mathbf{Q}_3 \mathbf{h}_k = \begin{bmatrix} k^2 \\ k \\ 1 \\ \mathbf{0} \end{bmatrix}, \quad \mathbf{K}_3 \mathbf{h}_\ell = \begin{bmatrix} -1/2 \\ (1-\ell)/2 \\ 1-\ell/2 \\ \mathbf{0} \end{bmatrix}, \quad \mathbf{V}_3 \mathbf{h}_j^x = \begin{bmatrix} \mathbf{0}_{d+1} \\ -\eta\lambda\mathbf{w}_j \\ \mathbf{0}_{D_{\mathrm{hid}}-2d-1} \end{bmatrix}, \quad \mathbf{V}_3 \mathbf{h}_j^y = \mathbf{0}_{D_{\mathrm{hid}}}.$$

For any $\ell \leq k$, we have

$$\sigma(\langle \mathbf{Q}_3 \mathbf{h}_k, \mathbf{K}_3 \mathbf{h}_\ell \rangle) = \sigma\big(-k^2/2 + k(1-\ell)/2 + 1 - \ell/2\big) = \frac{k-1}{2}\sigma(-k+\ell+1) = \frac{k-1}{2}\mathbb{1}\{\ell = k\}.$$

Therefore, for query token $k = 2i - 1$, the attention head outputs

$$\sum_{\ell=1}^{k} \frac{1}{k}\sigma(\langle \mathbf{Q}_3 \mathbf{h}_k, \mathbf{K}_3 \mathbf{h}_\ell \rangle)\mathbf{V}_m \mathbf{h}_\ell = \sum_{\ell=1}^{k} \frac{1}{k} \cdot \frac{k-1}{2}\mathbb{1}\{\ell = k\} \cdot \mathbf{V}_m \mathbf{h}_\ell$$

$$= \frac{k-1}{2k} \cdot \mathbf{V}_m \mathbf{h}_k = \frac{i-1}{2i-1} \cdot \mathbf{V}_m \mathbf{h}_i^x = \frac{i-1}{2i-1} \cdot [\mathbf{0}_{d+1}; -\eta\lambda\mathbf{w}_i; \mathbf{0}_{D_{\mathrm{hid}}-2d-1}]. \tag{15}$$

It is straightforward to see that the same attention head at query token $k = 2i$ outputs $\mathbf{0}_{D_{\mathrm{hid}}}$, as the value vector $\mathbf{V}_3 \mathbf{h}_k = \mathbf{V}_3 \mathbf{h}_i^y$ is zero.

Combining (14) and (15), letting the full attention layer $\boldsymbol{\theta} := \{(\mathbf{Q}_m, \mathbf{K}_m, \mathbf{V}_m)\}_{m=1,2,3}$, we have $\mathrm{Attn}_{\boldsymbol{\theta}}(\mathbf{H}) = \widetilde{\mathbf{H}}$, where for all $i \in [N]$,

$$\widetilde{\mathbf{h}}_i^x = \widetilde{\mathbf{h}}_{2i-1} = \mathbf{h}_{2i-1} + \sum_{m=1}^{3}\sum_{\ell=1}^{2i-1} \frac{1}{2i-1}\sigma(\langle \mathbf{Q}_m \mathbf{h}_{2i-1}, \mathbf{K}_m \mathbf{h}_\ell \rangle) \cdot \mathbf{V}_m \mathbf{h}_\ell$$

$$= \begin{bmatrix} \mathbf{x}_i \\ 0 \\ \mathbf{w}_i \\ * \end{bmatrix} + \frac{i-1}{2i-1}\begin{bmatrix} \mathbf{0}_{d+1} \\ -\eta\big(\nabla\widehat{L}_{i-1}(\mathbf{w}_i) + \lambda\mathbf{w}_i\big) \\ \mathbf{0}_{D_{\mathrm{hid}}-2d-1} \end{bmatrix} = \begin{bmatrix} \mathbf{x}_i \\ 0 \\ \mathbf{w}_i - \eta_i \nabla\widehat{L}_{i-1}^\lambda(\mathbf{w}_i) \\ * \end{bmatrix},$$

where $\eta_i := \frac{i-1}{2i-1}\mathbf{w}_i$, and $\widetilde{\mathbf{h}}_i^y = \mathbf{h}_i^y$. This finishes the proof. $\qquad\square$

**Theorem B.5** (In-context ridge regression by decoder-only transformer). *For any* $\lambda \geq 0$, $B_x, B_w, B_y > 0$ *with* $\kappa := 1 + B_x^2/\lambda$, *and* $\varepsilon < B_x B_w/2$, *let* $D_{\mathrm{hid}} \geq 2d + 10$, *then there exists an $L$-layer transformer $\mathrm{TF}_{\boldsymbol{\theta}}$ with $M = 3$ heads and hidden dimension $D_{\mathrm{hid}}$, where*

$$L = \lceil 3\kappa \log(B_x B_w/(2\varepsilon)) \rceil + 2, \tag{16}$$

*such that the following holds. On any input matrix $\mathbf{H}$ of form (1) such that problem (13) has bounded inputs and solution: for all $i \in [N]$*

$$\|\mathbf{x}_i\|_2 \leq B_x, \quad |y_i| \leq B_y, \quad \big\|\widehat{\mathbf{w}}_i^\lambda\big\|_2 \leq B_w/2, \tag{17}$$

*$\mathrm{TF}_{\boldsymbol{\theta}}$ approximately implements the ridge regression algorithm (minimizer of risk (13)) at every token $i \in [N]$: The prediction $\widehat{y}_i := [\mathrm{TF}_{\boldsymbol{\theta}}(\mathbf{H})]_{d+1,2i-1}$ satisfies*

$$\big|\widehat{y}_i - \langle \widehat{\mathbf{w}}_i^\lambda, \mathbf{x}_i \rangle\big| \leq \varepsilon. \tag{18}$$

*Proof.* The proof consists of two steps.

**Step 1** We analyze the convergence rate of gradient descent on $\widehat{L}_{i-1}^\lambda$ simultaneously for all $2 \leq i \leq N$, each with learning rate $\eta_i = \frac{i-1}{2i-1}\eta$ as implemented in Proposition B.4.

Fix $2 \leq i \leq N$. Consider the ridge risk $\widehat{L}_{i-1}^\lambda$ defined in (13), which is a convex quadratic function that is $\lambda$-strongly convex and $\lambda_{\max}\big(\mathbf{X}_{i-1}^\top \mathbf{X}_{i-1}/(i-1)\big) + \lambda \leq B_x^2 + \lambda =: \beta$ smooth over $\mathbb{R}^d$. Recall $\kappa = \beta/\lambda = 1 + B_x^2/\lambda$.

Consider the following gradient descent algorithm on $\widehat{L}_{i-1}^\lambda$: Initialize $\mathbf{w}_i^0 := \mathbf{0}$, and for every $t \geq 0$

$$\mathbf{w}_i^{t+1} = \mathbf{w}_i^t - \eta_i \nabla \widehat{L}_{i-1}^\lambda(\mathbf{w}_i^t), \tag{19}$$

with $\eta_i = \frac{i-1}{2i-1}\eta$. Taking $\eta := 2/\beta$, we have $\eta_i \in [2/(3\beta), 1/\beta]$, and thus $\eta_i \lambda \in [2/(3\kappa), 1/\kappa]$.

By standard convergence results for strongly convex and smooth functions (Proposition A.1), we have for all $t \geq 1$ that

$$\left\| \mathbf{w}_i^t - \widehat{\mathbf{w}}_i^\lambda \right\|_2^2 \leq \exp\left(-\eta_i \lambda t\right) \left\| \mathbf{w}_i^0 - \widehat{\mathbf{w}}_i^\lambda \right\|_2^2 = \exp\left(-\eta_i \lambda t\right) \left\| \widehat{\mathbf{w}}_i^\lambda \right\|_2^2.$$

Further, taking the number of steps as

$$T := \left\lceil 3\kappa \log\left(\frac{B_x B_w}{2\varepsilon}\right) \right\rceil$$

so that $\eta_i \lambda T/2 \geq 2/(3\kappa) \cdot 3\kappa \log(B_x B_w/(2\varepsilon))/2 = \log(B_x B_w/(2\varepsilon))$, we have

$$\left\| \mathbf{w}_i^T - \widehat{\mathbf{w}}_i^\lambda \right\|_2 \leq \exp\left(-\eta_i \lambda T/2\right) \left\| \widehat{\mathbf{w}}_i^\lambda \right\|_2 \leq \frac{2\varepsilon}{B_x B_w} \cdot \frac{B_w}{2} \leq \frac{\varepsilon}{B_x}. \tag{20}$$

**Step 2** We construct a $(T+2)$-layer transformer $\mathrm{TF}_{\boldsymbol{\theta}}$ by concatenating the copying layer in Lemma B.1, $T$ identical gradient descent layers as constructed in Proposition B.4, and the linear prediction layer in Lemma B.2. Note that the transformer is attention only (all MLP layers being zero), and the number of heads within all layers is at most 3.

The copying layer ensures that the output format is compatible with the input format required in Proposition B.4, which in turn ensures that the $T$ gradient descent layers implement (19) simultaneously for all $1 \leq i \leq N$ ($\mathbf{w}_1^T := \mathbf{0}$ is not updated at token $i = 1$). Therefore, the final linear prediction layer ensures that, the output matrix $\widetilde{\mathbf{H}} := \mathrm{TF}_{\boldsymbol{\theta}}(\mathbf{H})$ contains the following prediction at every $i \in [N]$:

$$\widehat{y}_i := [\widetilde{\mathbf{h}}_i^x]_{d+1} = \left\langle \mathbf{w}_i^T, \mathbf{x}_i \right\rangle,$$

which satisfies

$$\left| \widehat{y}_i - \left\langle \widehat{\mathbf{w}}_i^\lambda, \mathbf{x}_i \right\rangle \right| = \left| \left\langle \mathbf{w}_i^T - \widehat{\mathbf{w}}_i^\lambda, \mathbf{x}_i \right\rangle \right| \leq (\varepsilon/B_x) \cdot B_x = \varepsilon.$$

This finishes the proof. $\qquad\square$

### B.3 PROOF OF THEOREM 1

The result follows directly by concatenating the following two transformer constructions:

- The MLP implementation module in Lemma B.3, which has $(L+1)$-layers, 5 heads, and transforms every $\mathbf{x}_i$ to $\Phi^\star(\mathbf{x}_i)$ to give output matrix (4);
- The in-context ridge regression module in Theorem B.5 (with inputs being $\{\Phi^\star(\mathbf{x}_i)\}$ instead of $\mathbf{x}_i$) which has $\mathcal{O}(\kappa \log(B_\Phi B_w/\varepsilon))$ layers, 3 heads, and outputs prediction $\widehat{y}_i := [\widetilde{\mathbf{h}}_i^x]_{D+1}$ where $|\widehat{y}_i - \langle \Phi^\star(\mathbf{x}_i), \widehat{\mathbf{w}}_i^{\Phi^\star, \lambda}\rangle| \leq \varepsilon$, where $\widehat{\mathbf{w}}_i^{\Phi^\star, \lambda}$ is the ($\Phi^\star$-Ridge) predictor.

Claim (4) can be seen by concatenating the $(L+1)$-layer MLP module with the first layer in the ridge regression module (Theorem B.5), which copies the $\Phi^\star(\mathbf{x}_i)$ in each $x$ token to the same location in the succeeding $y$ token.

Further, the hidden dimension requirements are $D_{\mathrm{hid}} \geq \max\{2D, D+d+10\}$ for the first module and $D_{\mathrm{hid}} \geq 2D + 10$ for the second module, which is satisfied at our precondition $D_{\mathrm{hid}} = 2D + d + 10$. This finishes the proof. $\qquad\square$

### C PROOFS FOR SECTION 4.2

Recall our input format (6) for the dynamical system setting:

$$\mathbf{H} := \begin{bmatrix} \mathbf{x}_1 & \cdots & \mathbf{x}_N \\ \mathbf{p}_1 & \cdots & \mathbf{p}_N \end{bmatrix} \in \mathbb{R}^{D_{\mathrm{hid}} \times N},$$

our choice of the positional encoding vectors $\mathbf{p}_i = [\mathbf{0}_{D_{\mathrm{hid}}-d-4}; 1; i; i^2; i^3]$ for all $i \in [N]$, and that we understand $\mathbf{x}_i := \mathbf{0}$ for all $i \leq 0$.

## C.1 USEFUL TRANSFORMER CONSTRUCTIONS

**Lemma C.1** (Copying for dynamical systems). *Suppose $D_{\mathrm{hid}} \geq kd + 4$. For any $k \in [N]$, there exists a $(k+1)$-head attention layer $\boldsymbol{\theta} = \{(\mathbf{Q}_m, \mathbf{K}_m, \mathbf{V}_m)\}_{m \in [k+1]} \subset \mathbb{R}^{D_{\mathrm{hid}} \times D_{\mathrm{hid}}}$ such that for every input $\mathbf{H}$ of the form (6), we have*

$$\widetilde{\mathbf{H}} = \mathrm{Attn}_{\boldsymbol{\theta}}(\mathbf{H}) = \begin{bmatrix} \mathbf{x}_{1-k+1} & \cdots & \mathbf{x}_{i-k+1} & \cdots & \mathbf{x}_{N-k+1} \\ | & & | & & | \\ \mathbf{x}_1 & \cdots & \mathbf{x}_i & \cdots & \mathbf{x}_N \\ \overline{\mathbf{p}}_1 & \cdots & \overline{\mathbf{p}}_i & \cdots & \overline{\mathbf{p}}_N \end{bmatrix} \in \mathbb{R}^{D_{\mathrm{hid}} \times N}, \qquad (21)$$

*where $\overline{\mathbf{p}}_i$ only differs from $\mathbf{p}_i$ in the dimension of the zero paddings. In words, $\mathrm{Attn}_{\boldsymbol{\theta}}$ copies the $k-1$ previous tokens $[\mathbf{x}_{i-k+1}; \ldots; \mathbf{x}_{i-1}]$ onto the $i$-th token.*

*Proof.* For every $k' \in [k]$, we define an attention head $(\mathbf{Q}_{k'}, \mathbf{K}_{k'}, \mathbf{V}_{k'}) \subset \mathbb{R}^{D_{\mathrm{hid}} \times D_{\mathrm{hid}}}$ such that for all $j \leq i \in [N]$,

$$\begin{aligned} \mathbf{Q}_{k'} \mathbf{h}_i &= [i^3; i^2; i; \mathbf{0}_{D_{\mathrm{hid}}-3}], \\ \mathbf{K}_{k'} \mathbf{h}_j &= [-1; 2j + 2(k'-1); -j^2 + 2(k'-1)j + 1 - (k'-1')^2; \mathbf{0}_{D_{\mathrm{hid}}-3}], \\ \mathbf{V}_{k'} \mathbf{h}_j &= [\mathbf{0}_{(k-k')D}; \mathbf{x}_j; \mathbf{0}]. \end{aligned}$$

Note that

$$\begin{aligned} \sigma(\langle \mathbf{Q}_{k'} \mathbf{h}_i, \mathbf{K}_{k'} \mathbf{h}_j \rangle) &= \sigma\big(-i^3 + 2i^2 j + 2(k'-1)i^2 - ij^2 + 2ij(k'-1) + i - i(k'-1)^2\big) \\ &= i\sigma\big(1 - (j - i + k' - 1)^2\big) = i\mathbb{1}\{j = i - k' + 1\}. \end{aligned}$$

Therefore, at output token $i \in [N]$, this attention head gives

$$\frac{1}{i} \sum_{j=1}^{i} \sigma(\langle \mathbf{Q}_{k'} \mathbf{h}_i, \mathbf{K}_{k'} \mathbf{h}_j \rangle) \mathbf{V}_{k'} \mathbf{h}_j = \frac{1}{i} \cdot i \cdot \mathbf{V}_{k'} \mathbf{h}_{i-k'+1} = [\mathbf{0}_{(k-k')D}; \mathbf{x}_{i-k'+1}; \mathbf{0}]$$

when $i - k' + 1 \geq 1$, and zero otherwise. Combining all $k$ heads, and defining one more head $(\mathbf{Q}_{k+1}, \mathbf{K}_{k+1}, \mathbf{V}_{k+1})$ to "remove" $\mathbf{x}_i$ at its original location (similar as in the proof of Lemma B.3), we have

$$\sum_{m=1}^{k+1} \frac{1}{i} \sum_{j=1}^{i} \sigma(\langle \mathbf{Q}_{k'} \mathbf{h}_i, \mathbf{K}_{k'} \mathbf{h}_j \rangle) \mathbf{V}_{k'} \mathbf{h}_j = \begin{bmatrix} \mathbf{x}_{i-k+1} - \mathbf{x}_i \\ \mathbf{x}_{i-(k-1)+1} \\ | \\ \mathbf{x}_i \\ \mathbf{0} \end{bmatrix}.$$

By the residual structure of an attention layer, we have

$$[\mathrm{Attn}_{\boldsymbol{\theta}}(\mathbf{H})]_i = \begin{bmatrix} \mathbf{x}_i \\ \mathbf{p}_i \end{bmatrix} + \begin{bmatrix} \mathbf{x}_{i-k+1} - \mathbf{x}_i \\ \mathbf{x}_{i-(k-1)+1} \\ | \\ \mathbf{x}_i \\ \mathbf{0} \end{bmatrix} = \begin{bmatrix} \mathbf{x}_{i-k+1} \\ \mathbf{x}_{i-(k-1)+1} \\ | \\ \mathbf{x}_i \\ \overline{\mathbf{p}}_i \end{bmatrix}.$$

(The precondition $D_{\mathrm{hid}} \geq D + 4$ guarantees that the $x$ entries would not interfere with the non-zero entries within $\mathbf{p}_i$.) This is the desired result. $\square$

**Lemma C.2** (Implementing MLP representation for dynamical systems). *Fix any MLP representation function $\Phi^\star : \mathbb{R}^{kd} \to \mathbb{R}^D$ of the form (3), suppose $D_{\mathrm{hid}} \geq 2(k+1)d + 3D + 2d + 5$. Then there exists a module MLP-(Attention-MLP-...-Attention-MLP) with $L+1$ (Attention-MLP) blocks (i.e. transformer layers) and $5$ heads in each attention layer (this is equivalent to an $(L+2)$-layer transformer without the initial attention layer) that implements $\Phi^\star$ in the following fashion: For any input $\mathbf{H}$ of form*

$$\mathbf{H} = \begin{bmatrix} \overline{\mathbf{x}}_1 & \cdots & \overline{\mathbf{x}}_N \\ \overline{\mathbf{p}}_1 & \cdots & \overline{\mathbf{p}}_N \end{bmatrix}$$

*where we recall* $\overline{\mathbf{x}}_i = [\mathbf{x}_{i-k+1}; \ldots; \mathbf{x}_i] \in \mathbb{R}^{kd}$, *the following holds. The first MLP layer outputs*

$$\mathrm{MLP}^{(1)}(\mathbf{H}) = \begin{bmatrix} \sigma_\rho(\mathbf{B}_1^\star \overline{\mathbf{x}}_1) & \ldots & \sigma_\rho(\mathbf{B}_1^\star \overline{\mathbf{x}}_i) \\ \mathbf{x}_1 & \ldots & \mathbf{x}_i \\ \overline{\mathbf{p}}_1' & \ldots & \overline{\mathbf{p}}_i' \end{bmatrix}.$$

*The full transformer outputs*

$$\widetilde{\mathbf{H}} = \mathrm{TF}_{\boldsymbol{\theta}}(\mathbf{H}) = \begin{bmatrix} \Phi^\star(\overline{\mathbf{x}}_1) & \Phi^\star(\overline{\mathbf{x}}_2) & \ldots & \Phi^\star(\overline{\mathbf{x}}_i) \\ \mathbf{0}_d & \mathbf{0}_d & \ldots & \mathbf{0}_d \\ \mathbf{0}_D & \Phi^\star(\overline{\mathbf{x}}_1) & \ldots & \Phi^\star(\overline{\mathbf{x}}_{i-1}) \\ \mathbf{x}_1 & \mathbf{x}_2 & \ldots & \mathbf{x}_i \\ \widetilde{\mathbf{p}}_1 & \widetilde{\mathbf{p}}_2 & \ldots & \widetilde{\mathbf{p}}_i \end{bmatrix}. \tag{22}$$

*where* $\widetilde{\mathbf{p}}_i, \widetilde{\mathbf{p}}_i$ *differs from* $\overline{\mathbf{p}}_i, \overline{\mathbf{p}}_i$ *only in the dimension of their zero paddings.*

*Proof.* We first construct the first MLP layer. Consider any input token $\mathbf{h}_i = [\overline{\mathbf{x}}_i; \mathbf{p}_i]$. Define matrices $\mathbf{W}_1, \mathbf{W}_2 \in \mathbb{R}^{D_{\mathrm{hid}} \times D_{\mathrm{hid}}}$ such that (below $\pm \mathbf{u} := [\mathbf{u}; -\mathbf{u}]$)

$$\mathbf{W}_1 \mathbf{h}_i = \begin{bmatrix} \pm \mathbf{B}_1^\star \overline{\mathbf{x}}_i \\ \pm \mathbf{x}_i \\ \pm \overline{\mathbf{x}}_i \\ \mathbf{0} \end{bmatrix}, \quad \sigma(\mathbf{W}_1 \mathbf{h}_i) = \begin{bmatrix} \sigma(\pm \mathbf{B}_1^\star \mathbf{x}_i) \\ \sigma(\pm \mathbf{x}_i) \\ \sigma(\pm \overline{\mathbf{x}}_i) \\ \mathbf{0} \end{bmatrix},$$

$$\mathbf{W}_2 \sigma(\mathbf{W}_1 \mathbf{h}_i) = \begin{bmatrix} \sigma(\mathbf{B}_1^\star \overline{\mathbf{x}}_i) - \rho \sigma(-\mathbf{B}_1^\star \overline{\mathbf{x}}_i) \\ \mathbf{0} \end{bmatrix} + \begin{bmatrix} -\sigma(\overline{\mathbf{x}}_i) + \sigma(-\overline{\mathbf{x}}_i) \\ \mathbf{0} \end{bmatrix} + \begin{bmatrix} \mathbf{0}_D \\ \sigma(\mathbf{x}_i) - \sigma(-\mathbf{x}_i) \\ \mathbf{0} \end{bmatrix}.$$

Therefore, the MLP layer $(\mathbf{W}_1, \mathbf{W}_2)$ outputs

$$\overline{\mathbf{h}}_i := [\mathrm{MLP}_{\mathbf{W}_1, \mathbf{W}_2}(\mathbf{H})]_i = \mathbf{h}_i + \mathbf{W}_2 \sigma(\mathbf{W}_1 \mathbf{h}_i) = \begin{bmatrix} \sigma_\rho(\mathbf{B}_1^\star \overline{\mathbf{x}}_i) \\ \mathbf{x}_i \\ \overline{\mathbf{P}}_i \end{bmatrix}. \tag{23}$$

The requirement for $D_{\mathrm{hid}}$ above is $D_{\mathrm{hid}} \geq \max \{2D + 2(k+1)d, D + d + 5\}$.

The rest of the proof follows by repeating the proof of Lemma B.3 (skipping the first (MLP-Attention) block), with the following modifications:

- Save the $\mathbf{x}_i \in \overline{\mathbf{x}}_i$ location within each token, and move it into the $(2D + d + 1 : 2D + 2d)$ block in the final layer (instead of moving the label $y_i$ in Lemma B.3); this takes the same number (at most 2) of attention heads in every layer, same as in Lemma B.3.

- Append one more copying layer with a single attention head (similar as the construction in Lemma C.1) to copy each $\Phi^\star(\overline{\mathbf{x}}_i)$ to the $(D + d + 1 : 2D + d)$ block of the next token.

The above module has structure $(L - 1) \times$(MLP-Attention), followed by a single attention layer which can be rewritten as an MLP-Attention-MLP module with identity MLP layers. Altogether, the module has an MLP-$L \times$(Attention-MLP) structure. The max number of attention heads within the above module is 5. The required hidden dimension here is $D_{\mathrm{hid}} \geq \max \{kd + 4, 2D + d + \max \{D, d\} + 5\}$, with $D_{\mathrm{hid}} \geq \max \{kd, 3D + 2d\} + 5$ being a sufficient condition.

Combining the above two parts, a sufficient condition for $D_{\mathrm{hid}}$ is $D_{\mathrm{hid}} \geq 2(k+1)d + 3D + 2d + 5$, as assumed in the precondition. This finishes the proof. $\square$

Consider the following multi-output ridge regression problem:

$$\widehat{\mathbf{W}}_i^\lambda := \underset{\mathbf{W} \in \mathbb{R}^{D \times d}}{\arg \min} \frac{1}{2(i-1)} \sum_{j=1}^{i-1} \left\| \mathbf{W}^\top \mathbf{x}_j - \mathbf{y}_j \right\|_2^2 + \frac{\lambda}{2} \left\| \mathbf{W} \right\|_{\mathsf{Fr}}^2. \tag{24}$$

**Theorem C.3** (In-context multi-output ridge regression with alternative input structure). *For any* $\lambda \geq 0$, $B_x, B_w, B_y > 0$ *with* $\kappa := 1 + B_x^2/\lambda$, *and* $\varepsilon < B_x B_w/2$, *let* $D_{\mathrm{hid}} \geq Dd + 2(D + d) + 5$, *then there exists an $L$-layer transformer* $\mathrm{TF}_{\boldsymbol{\theta}}$ *with* $M = 3d$ *heads and hidden dimension* $D_{\mathrm{hid}}$, *where*

$$L = \mathcal{O}(\kappa \log(B_x B_w/(\varepsilon))) \tag{25}$$

*such that the following holds. On any input matrix*

$$\mathbf{H} = \begin{bmatrix} \mathbf{x}_1 & \mathbf{x}_2 & \dots & \mathbf{x}_N \\ \mathbf{0}_d & \mathbf{0}_d & \dots & \mathbf{0}_d \\ \mathbf{0}_D & \mathbf{x}_1 & \dots & \mathbf{x}_{N-1} \\ \mathbf{0}_d & \mathbf{y}_1 & \dots & \mathbf{y}_{N-1} \\ \mathbf{p}_1 & \mathbf{p}_2 & \dots & \mathbf{p}_N \end{bmatrix}$$

*(where $\mathbf{x}_i \in \mathbb{R}^D$, $\mathbf{y}_i \in \mathbb{R}^d$) such that problem (24) has bounded inputs and solution: for all $i \in [N]$*

$$\|\mathbf{x}_i\|_2 \leq B_x, \quad \|\mathbf{y}_i\|_{\infty} \leq B_y, \quad \|\widehat{\mathbf{W}}_i^{\lambda}\|_{2,\infty} \leq B_w/2, \tag{26}$$

$\mathrm{TF}_{\boldsymbol{\theta}}$ *approximately implements the ridge regression algorithm (24) at every token $i \in [N]$: The prediction $\widehat{\mathbf{y}}_i := [\mathrm{TF}_{\boldsymbol{\theta}}(\mathbf{H})]_{(D+1):(D+d),i}$ satisfies*

$$\left\| \widehat{\mathbf{y}}_i - (\widehat{\mathbf{W}}_i^{\lambda})^{\top} \mathbf{x}_i \right\|_{\infty} \leq \varepsilon. \tag{27}$$

*Proof.* Observe that the multi-output ridge regression problem (24) is equivalent to $d$ separable single-output ridge regression problems, one for each output dimension. Therefore, the proof follows by directly repeating the same analysis as in Theorem B.5, with the adaptation that

- Omit the copying layer since each token already admits the previous (input, label) pair;

- Use a $\mathcal{O}(\kappa \log(B_x B_w/(\varepsilon)))$-layer transformer with $3d$ heads to perform $d$ parallel ridge regression problems (each with 3 heads), using in-context gradient descent (Proposition B.4) as the internal optimization algorithm, and with slightly different input structures that can be still accommodated by using relu to implement the indicators. Further, by the precondition (26) and $D_{\mathrm{hid}} - 2(D + d) - 5 \geq Dd$, we have enough empty space to store the $\mathbf{W}_i^t \in \mathbb{R}^{D \times d}$ within the zero-paddings in $\mathbf{p}_i$.

- Use a single-attention layer with $d$ parallel linear prediction heads (Lemma B.2), one for each $j \in [d]$, to write prediction $(\widehat{\mathbf{y}}_i)_j$ into location $(i, D + j)$ with $|(\widehat{\mathbf{y}}_i)_j - \langle (\widehat{\mathbf{W}}_i^{\lambda})_j, \mathbf{x}_i \rangle| \leq \varepsilon$. Therefore,

$$\left\| \widehat{\mathbf{y}}_i - (\widehat{\mathbf{W}}_i^{\lambda})^{\top} \mathbf{x}_i \right\|_{\infty} = \max_{j \in [d]} \left| (\widehat{\mathbf{y}}_i)_j - \left\langle (\widehat{\mathbf{W}}_i^{\lambda})_j, \mathbf{x}_i \right\rangle \right| \leq \varepsilon.$$

This finishes the proof. $\qquad\square$

## C.2 PROOF OF THEOREM 2

*Proof of Theorem 2.* The proof is similar as that of Theorem 1. The result follows directly by concatenating the following three transformer modules:

- The copying layer in Lemma C.1, which transforms the input to format (21), and thus verifies claim (7).

- The MLP representation module in Lemma C.2, which transforms (21) to (22). Together with the above single attention layer, the module is now an $(L + 1)$-layer transformer with 5 heads. Claim (8) follows by the intermediate output (23) within the proof of Lemma C.2.

- The in-context multi-output ridge regression construction in Theorem C.3 (with inputs being $\{\Phi^{\star}(\overline{\mathbf{x}}_i)\}$ and labels being $\{\mathbf{x}_{i+1}\}$). This TF has $\mathcal{O}(\kappa \log(B_{\Phi} B_w/\varepsilon))$ layers, and $3d$ heads. It takes in input of format (22), and outputs prediction $\widehat{\mathbf{y}}_i := [\widetilde{\mathbf{h}}_i]_{D+1:D+d}$ where $\|\widehat{\mathbf{y}}_i - (\widehat{\mathbf{W}}_i^{\Phi^{\star}, \lambda})^{\top} \Phi^{\star}(\overline{\mathbf{x}}_i)\|_{\infty} \leq \varepsilon$, where $\widehat{\mathbf{W}}_i^{\Phi^{\star}, \lambda}$ is the ($\Phi^{\star}$-Ridge-Dyn) predictor.

The resulting transformer has $\max\{3d, 5\}$ heads, and the hidden dimension requirement is $D_{\text{hid}} \geq \max\{kd + 5, 2(k+1)d + 3D + 2d + 5, Dd + 2(D + d) + 5\}$. A sufficient condition is $D_{\text{hid}} = \max\{2(k+1), D\}d + 3(D + d) + 5$, as assumed in the precondition. This finishes the proof. $\qquad\square$

## D    DETAILS FOR EXPERIMENTS

**Architecture and training details** We train a 12-layer decoder model in GPT-2 family with 8 heads and hidden dimension $D_{\text{hid}} = 256$, with positional encoding. We use linear read-in and read-out layer before and after the transformers respectively, both applying a same affine transform to all tokens in the sequence and are trainable. The read-in layer maps any input vector to a $D_{\text{hid}}$-dimensional hidden state, and the read-out layer maps a $D_{\text{hid}}$-dimensional hidden state to a 1-dimensional scalar for model (2) and to a $d$-dimensional scalar for model (5).

Under the in-context learning with representation setting, we first generate and fix the representation $\Phi^\star$. For a single ICL instance, We generate new coefficients $\mathbf{w}$ and $N$ training examples $\{(\mathbf{x}_i, y_i)\}_{i \in [N]}$ and test input $(\mathbf{x}_{N+1}, y_{N+1})$. Before feeding into transformer, we re-format the sequence to $\mathbf{H}_{\text{ICL-rep}}$, as shown in equation (28).

$$\mathbf{H}_{\text{ICL-rep}} = \left[\mathbf{x}_1, \begin{bmatrix} y_1 \\ \mathbf{0}_{d-1} \end{bmatrix}, \ldots, \mathbf{x}_N, \begin{bmatrix} y_N \\ \mathbf{0}_{d-1} \end{bmatrix}\right] \in \mathbb{R}^{d \times 2N} \tag{28}$$

We use the use the Adam optimizer with a fixed learning rate $10^{-4}$, which works well for all experiments. We train the model for $300K$ steps, where each step consists of a (fresh) minibatch with batch size 64 for single representation experiments, except for the mixture settings in Appendix E where we train for $150K$ iterations, each containing $K$ batches one for each task.

Under ICL dynamic system setting, for a single ICL instance, we don't need to reformat the input sequence. We feed the original sequence $\mathbf{H}_{\text{Dynamic}} = [\mathbf{x}_1, \ldots, \mathbf{x}_N] \in \mathbb{R}^{d \times N}$ to transformer.

**Generating representations:**    Denote the column-wise orthogonal matrices with size $r \times s$ ($r \geq s$) as $\mathcal{O}(r, s)$. We sample $\text{Unif}[\mathcal{O}(r, s)]$ through generating an $r \times r$ matrix, getting its QR decomposition, then taking the first $s$ columns of the $Q$ matrix. To generate $L$ layers MLP with hidden dimension $D$. We first generate $L$ weight matrices $\mathbf{B}_\ell$, $\ell = 1, \ldots, L$, with $\mathbf{B}_1 \sim \text{Unif}[\mathcal{O}(d, D)]$ (if $d < D$, we sample $\mathbf{B}_1^\top \sim \text{Unif}[\mathcal{O}(d, D)]$) and the rest independently from $\text{Unif}[\mathcal{O}(D, D)]$. After calculating $\text{MLP}(\mathbf{x})$, we self-normalize it with $\|\text{MLP}(\mathbf{x})\|_2$.

All our plots show one-standard-deviation error bars, though some of those are not too visible.

### D.1    DETAILS FOR LINEAR PROBING

Denote the $\ell-$th hidden state of transformers as

$$\mathbf{H}^{(\ell)} = \left[\mathbf{h}_1^{x,(\ell)}, \mathbf{h}_1^{y,(\ell)}, \ldots, \mathbf{h}_N^{x,(\ell)}, \mathbf{h}_N^{y,(\ell)}\right] \in \mathbb{R}^{D_{\text{hid}}, 2N} \quad \text{for} \quad \ell \in [12].$$

Denote the probing target as $g\left(\{\mathbf{x}_j, y_j\}_{j \in [i]}\right) \in \mathbb{R}^{d_{\text{probe}}}$ for $i \in [N]$. Denote the linear probing parameter as $\mathbf{w}^{x,(\ell)}$ and $\mathbf{w}^{y,(\ell)}$ that belong to $\mathbb{R}^{D_{\text{hid}} \times d_{\text{probe}}}$. Denote the best linear probing model as

$$\mathbf{w}_\star^{x,(\ell)} = \underset{\mathbf{w}^{x,(\ell)}}{\arg\min} \, \mathbb{E}\Big[\sum_{i=1}^N \Big\{\left(\mathbf{w}^{x,\ell}\right)^\top \mathbf{h}_i^{x,(\ell)} - g\left(\{\mathbf{x}_j, y_j\}_{j \in [i]}\right)\Big\}^2\Big] \quad \text{and}$$

$$\mathbf{w}_\star^{y,(\ell)} = \underset{\mathbf{w}^{y,(\ell)}}{\min} \, \mathbb{E}\Big[\sum_{i=1}^N \Big\{\left(\mathbf{w}^{y,\ell}\right)^\top \mathbf{h}_i^{y,(\ell)} - g\left(\{\mathbf{x}_j, y_j\}_{j \in [i]}\right)\Big\}^2\Big].$$

To find them, we generate 2560 ICL input sequences with length $N$, and obtain 12 hidden states for each input sequences. We leave 256 sequences as test sample and use the remaining samples to estimate $\mathbf{w}_\star^{x,(\ell)}$ and $\mathbf{w}_\star^{y,(\ell)}$ for each $\ell$ with ordinary least squares. We use the mean squared error to measure the probe errors. In specific, define

$$\text{Probe Error}_i^{x,(\ell)}(g) = \mathbb{E}\Big[\Big\{\left(\mathbf{w}_\star^{x,\ell}\right)^\top \mathbf{h}_i^{x,(\ell)} - g\left(\{\mathbf{x}_j, y_j\}_{j \in [i]}\right)\Big\}^2\Big] \quad \text{with}$$

$$\text{Probe Error}^{x,(\ell)}(g) \;=\; \frac{1}{N}\sum_{i=1}^{N}\text{Probe Error}_i^{x,(\ell)}(g), \quad \text{and}$$

$$\text{Probe Error}_i^{y,(\ell)}(g) \;=\; \mathbb{E}\left[\left\{\left(\mathbf{w}_\star^{x,\ell}\right)^\top\mathbf{h}_i^{x,(\ell)} - g\left(\{\mathbf{x}_j,y_j\}_{j\in[i]}\right)\right\}^2\right] \quad \text{with}$$

$$\text{Probe Error}^{y,(\ell)}(g) \;=\; \frac{1}{N}\sum_{i=1}^{N}\text{Probe Error}_i^{y,(\ell)}(g).$$

When $\ell = 0$, we let $\mathbf{h}_i^{x,(0)} = \mathbf{h}_i^{y,(0)} = \mathbf{x}_i$ as a control to the probe errors in the hidden layer. We normalize each probe error with $\mathbb{E}[\|g(\mathbf{x},y)\|_2^2]/d_{\text{probe}}$. We use the 256 leaved-out samples to estimate these errors. We replicate the above procedure for three times and take their mean to get the final probe errors.

## D.2 DETAILS FOR PASTING

From the single fixed representation settings above, we pick a trained transformer trained on the representation with $D = d = 20$ to avoid dimension mismatch between $\Phi^\star(\mathbf{x})$ and $\mathbf{x}$. We choose $L = 3$ and noise level $\sigma = 0.1$.

We change the data generating procedure of $y$ from Equation (2) to

$$y_i = \langle\mathbf{w},\mathbf{x}_i\rangle + \sigma z_i, \quad i \in [N], \tag{29}$$

which corresponds to a linear-ICL task. According to the results of probing Fig 3a, we conjecture that transformer use the first 4 layers to recover the representation, and implement in-context learning through the 5-th to the last layers. Therefore, we extract the $5 - 12$ layers as the transformer upper layers. Then paste them with three kinds of embeddings:

1. *Linear* embedding $\mathbf{W} \in \mathbb{R}^{D_{\text{hid}}\times(D+1)}$ with re-formatted input $\mathbf{H}_{\text{Linear}}$:

$$\mathbf{H}_{\text{Linear}} = \left[\begin{bmatrix}\mathbf{x}_1\\0\end{bmatrix},\begin{bmatrix}\mathbf{0}_D\\y_1\end{bmatrix},\dots,\begin{bmatrix}\mathbf{x}_N\\0\end{bmatrix},\begin{bmatrix}\mathbf{0}_D\\y_N\end{bmatrix}\right] \in \mathbb{R}^{D+1\times 2N}$$

2. *Linear copy* embedding $\mathbf{W} \in \mathbb{R}^{D_{\text{hid}}\times(D+1)}$ with re-formatted input $\mathbf{H}_{\text{copy}}$ that copies $\mathbf{x}_i$ to $y_i$ tokens in advance:

$$\mathbf{H}_{\text{copy}} = \left[\begin{bmatrix}\mathbf{x}_1\\0\end{bmatrix},\begin{bmatrix}\mathbf{x}_1\\y_1\end{bmatrix},\dots,\begin{bmatrix}\mathbf{x}_N\\0\end{bmatrix},\begin{bmatrix}\mathbf{x}_N\\y_N\end{bmatrix}\right] \in \mathbb{R}^{D+1\times 2N}$$

3. *Transformer* embedding TF using the same input format $\mathbf{H}_{\text{ICL}-\text{rep}}$ with normal settings, as shown in (28). We extract the 4-th layer of the GPT-2 model, its a complete transformer block with trainable layer norm. We use a linear read-in matrix to map $\mathbf{H}_{\text{ICL}-\text{rep}}$ to the $D_{\text{hid}}$-dimension hidden state, apply one block of transformer to it to get the TF embedding $\overline{\mathbf{H}} = \overline{\text{TF}}(\mathbf{H})$.

We apply the upper layers to the three embeddings, then use the original read-out matrix to get the prediction of $\widehat{y}_i$. For comparison, we also train a one-layer transformer using the input sequence $\mathbf{H}_{\text{ICL}-\text{rep}}$.

We use the same training objective as in (10). In the retraining process, we switch to task (29), fix the parameters of upper layers of the transformer, and only retrain the embedding model. The training methods are exact the same with the original transformer. We also find that using a random initialized transformer block or extracting the 4-th layer of the transformer don't make difference to the results.

## D.3 DIFFICULTY OF LINEAR ICL WITH A SINGLE-LAYER TRANSFORMER WITH SPECIFIC INPUT FORMAT

Recall the input format (1):

$$\mathbf{H} = \begin{bmatrix}\mathbf{x}_1 & \mathbf{0} & \dots & \mathbf{x}_N & \mathbf{0}\\0 & y_1 & \dots & 0 & y_N\\\mathbf{p}_1^x & \mathbf{p}_1^y & \dots & \mathbf{p}_N^x & \mathbf{p}_N^y\end{bmatrix} \in \mathbb{R}^{D_{\text{hid}}\times 2N}.$$

Here we heuristically argue that a single attention layer alone (the only part in a single-layer transformer that handles interaction across tokens) is unlikely to achieve good linear ICL performance on input format (1).

Consider a single attention head $(\mathbf{Q}, \mathbf{K}, \mathbf{V})$. As we wish the transformer to do ICL prediction at every token, the linear estimator $\mathbf{w}_i$ used to predict $\widehat{y}_i$ is likely best stored in the $\mathbf{x}_i$ token (the only token that can attend to all past data $\mathcal{D}_{i-1}$ and the current input $\mathbf{x}_i$). In this case, the attention layer needs to use the following (key, value) vectors to compute a good estimator $\mathbf{w}_i$ from the data $\mathcal{D}_{i-1}$:

$$\{\mathbf{V}\mathbf{h}_j^x, \mathbf{V}\mathbf{h}_j^x\}_{j \in [i]}, \quad \{\mathbf{V}\mathbf{h}_j^y, \mathbf{V}\mathbf{h}_j^y\}_{j \in [i-1]}.$$

However (apart from position information), $\mathbf{h}_j^x$ only contains $\mathbf{x}_j$, and $\mathbf{h}_j^y$ only contains $y_j$. Therefore, using the normalized ReLU activation as in Appendix B & C.2, it is unlikely that an attention layer can implement even simple ICL algorithms such as one step of gradient descent (von Oswald et al., 2022; Akyürek et al., 2022):

$$\mathbf{w}_i = \mathbf{w}_i^0 - \eta \frac{1}{i-1} \sum_{j \le i-1} \left( \langle \mathbf{w}_i^0, \mathbf{x}_j \rangle - y_j \right) \mathbf{x}_j,$$

which (importantly) involves term $-y_j \mathbf{x}_j$ that is unlikely to be implementable by the above attention, where each attention head at each key token can observe either $\mathbf{x}_j$ or $y_j$ but not both.

### D.4  REPRODUCIBILITY

Code for our experiments is provided at the following anonymous link[4].

## E  EXPERIMENTS ON MIXTURE OF MULTIPLE REPRESENTATIONS

We train transformers on a mixture of multiple ICL tasks, where each task admits a different representation function. This setting is a representation selection problem similar as the "algorithm selection" setting of Bai et al. (2023). In specific, let $K \ge 2$ denote the number of tasks. Given $j$, let

$$y_i = \langle \mathbf{w}, \Phi_j^\star(\mathbf{x}_i) \rangle + \sigma z_i, \quad z_i \sim \mathsf{N}(0, 1), \quad i \in [N], \quad \text{where}$$

$$\Phi_j^\star(\mathbf{x}) = \sigma^\star \left( \mathbf{B}_L^{\star,(j)} \sigma^{\star,(j)} \left( \mathbf{B}_{L-1}^\star \cdots \sigma^{\star,(j)} \left( \mathbf{B}_1^{\star,(j)} \mathbf{x} \right) \cdots \right) \right), \quad \mathbf{B}_1^{\star,(j)} \in \mathbb{R}^{D \times d}, \quad (\mathbf{B}_\ell^{\star,(j)})_{\ell=2}^L \subset \mathbb{R}^{D \times D}.$$

The distributions for $\mathbf{w}$, $\{\mathbf{x}_i\}_{i \in [N]}$, and $\{\mathbf{B}_L^{\star,(j)}\}$ are same with previous setting. We generate different $\Phi_j^\star$ for $j \in [K]$ independently. We choose $K \in \{3, 6\}$, $\sigma \in \{0, 0.1, 0.5\}$, $L = 3$, and noise $\sigma \in \{0, 0.1, 0.5\}$.

At each training step, we generate $K$ independent minibatches, with the $j-$th minimatch takes the representation $\Phi_j^\star$ to generate $\{y_i\}_{i \in [N]}$. Due to multiple minibatches, we shorten the number of total training steps to $150K$. The other training details are the same with fixed single representation setting.

**ICL performance**  We choose one representation $\Phi_1^\star$ from the representations that transformers are trained on. Figure 6a & Figure 6b report the test risk. We vary $K \in \{3, 6\}$ and noise level $\sigma \in \{0.1, 0.5\}$. We consider two baseline models.

1. *The Bayes optimal algorithm*: Note that the training distribution follows the Bayesian hierarchical model:

   $$j \sim \mathsf{Unif}([K]), \quad \mathbf{x}_i \sim \mathsf{N}(0, \mathbf{I}_d), \quad \mathbf{w} \sim \mathsf{N}(0, \tau^2 \mathbf{I}_d), \quad \text{and} \quad y_i \mid \mathbf{x}_i, j, \mathbf{w} \sim \mathsf{N}(\langle \mathbf{w}, \mathbf{x}_i \rangle, \sigma^2).$$

   This gives the Bayes optimal predictor

   $$\widehat{y}_i = \sum_{j=1}^K \eta_i^{(j)} \widehat{y}_i^{(j)}, \quad \text{with} \quad (\ldots, \eta_i^{(j)}, \ldots) = \mathsf{Softmax}\left\{ \left[ \ldots, \sum_{k=1}^i (y_k - \widehat{y}_k^{(j)})^2 / \sigma^2, \ldots \right] \right\}$$

   (30)

   with $\widehat{y}_i^{(j)}$ being ridge predictor with optimal $\lambda$ based on $\left\{ \left( \Phi_j^\star(\mathbf{x}_r), y_r \right) \right\}_{r \in [i-1]}$.

---

[4] https://anonymous.4open.science/r/tf-rep-icl

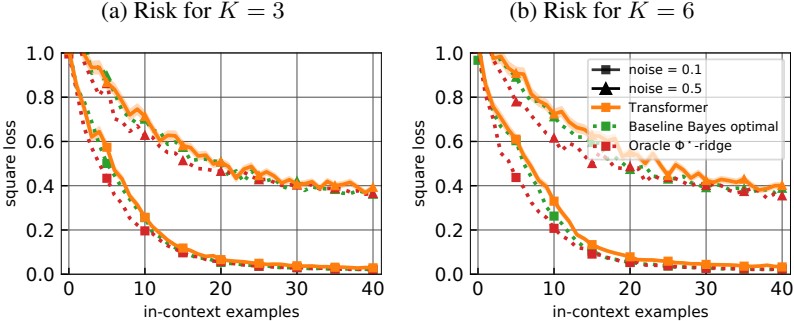

Figure 6: ICL risks for multiple representations setting. Dotted lines plot two baseline risks. **(a)** The transformer with lower risks is trained with $(K, L, D, \sigma) = (3, 3, 20, 0.1)$. The upper one is trained with $(K, L, D, \sigma) = (3, 3, 20, 0.5)$. **(b)** The two transformers are trained with $K = 6$ and same settings otherwise.

2. *The oracle ridge algorithm:* We use the ridge predictor $\widehat{y}_i^{(1)}$ based on $\{(\Phi_1^\star(\mathbf{x}_r), y_r)\}_{r \in [i-1]}$, which is the representation for test distribution. Note that this is an (improper) algorithm that relies on knowledge of the ground truth task.

Comparable to those trained on single fixed representation, transformers consistently match the Bayes-optimal ridge predictor. As expected, the oracle ridge algorithm is better than transformers and the Bayes optimal algorithm and transformers. Increasing number of tasks $K$ can slightly increase this gap. Increasing the noise level has the same effect on transformers and baseline algorithms.

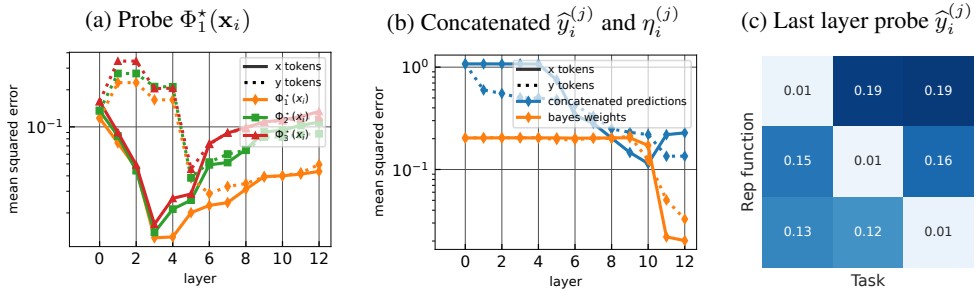

Figure 7: Probing errors for transformer trained with $(K, L, D, \sigma) = (3, 3, 20, 0.1)$. Dotted lines plot probing errors on $y$ tokens.

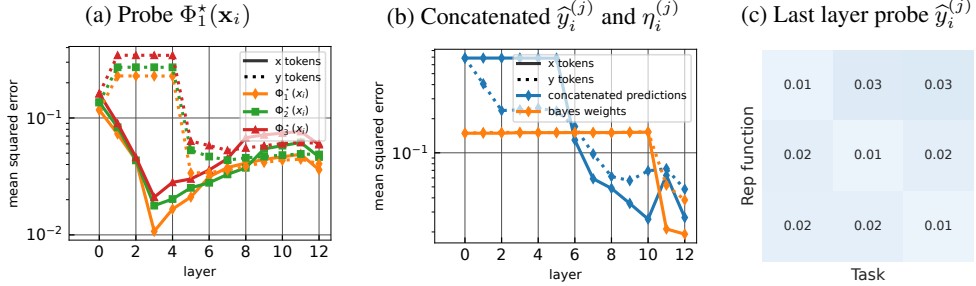

Figure 8: Probing errors for transformer trained with $\sigma = 0.5$

**Probe setup:** Similar to single fixed representation setting, we conduct linear probing experiments. We are wondering transformer implements the ICL-learning on representations with algorithm selections mechanism. We identify three sets of probing targets: $\Phi^\star(\mathbf{x}_i)$, $\widehat{y}^{(j)}$ and $\eta_i^{(j)}$. All of them are intermediate values to compute the Bayes optimal estimator (30). We generate different data for different probing targets:

1. To probe $\Phi^\star(\mathbf{x}_i)$ and $\widehat{y}^{(j)}$ for each $j$, we choose one from the representations that transformers are trained on, then train and test our linear probing model. This is the same with the training and testing methods for probing transformers trained on a single representation.

2. To probe choose concatenated probing targets $\mathbf{Y}_i = [\widehat{y}_i^{(1)}, \ldots, \widehat{y}_i^{(K)}]$ and $\mathbf{B}_i = [\eta_i^{(1)}, \ldots, \eta_i^{(K)}]$, we generate 2560 in-context sequences for each representation, and obtain $2560 \times K$ samples together. We use ordinary linear square on $2560 \times K - 256$ samples to get the linear probing models. Then test them on the remaining 256 samples to get the probing errors. We also repeat this process for three times and take means to get the final probing errors.

**Probe representations:** Take the transformer trained on $K = 3$ mixture representations with noise level $\sigma \in \{0.1, 0.5\}$. Figure 7 show the probing errors for $\sigma = 0.1$: Figure 7a reports the errors of probing $\Phi_j^\star$ $j \in [3]$, with probing models trained on task $\Phi_1^\star$. Echoing the results for transformers trained on single representation, the probing errors for each representations decrease through lower layers and increase through upper layers on $\mathbf{x}$ tokens. The probing errors on $y$ tokens drop after $\mathbf{x}$ tokens, which suggests a copy mechanism. Surprisingly, on $\mathbf{x}$-tokens, the probing errors for all representations attain their minimum at the 3-th layer, with transformers trained on single representation achieving their minimum on 4-th layer (compare with Figure 3a).

More importantly, for both $\mathbf{x}$ and $y$ tokens, the probing errors for each representation are similar through lower layers, but the probing errors for the true representation $\Phi_1^\star$ become the lowest through the upper layers. The gap between the probing errors increases. At the last layer, the probing error for the other representations go up to match the initial input.

**Probe intermediate values for computing Bayes optimal predictor:** Figure 7b shows the probing errors for concatenated ridge predictors $\widehat{y}_i^{(j)}$ and Bayes weights $\eta_i^{(j)}$, i.e., $\mathbf{Y}_i$ and $\mathbf{B}_i$. The probing errors for $\mathbf{Y}_i$ start dropping at the $4-$th layer, which suggest that transformer are implementing ICL using each representations. Probing errors for $\mathbf{B}_i$ have a sudden drop at the $10-$th layer. Figure 7c shows the probing errors for probing $\widehat{y}_i^{(j)}$. At $(j, k)$-th cell, we show the probing error of $\widehat{y}_i^{(j)}$ with probing models trained on $\Phi_k^\star$ at the $\mathbf{x}$ tokens of the last layer. The diagonal elements are much smaller than others. The results combined together suggest the possibility that transformer implements in-context learning with three representations and selects one at the $10-$th layer.

In comparison, Figure 8 shows results of probing the same targets for transformer under $\sigma = 0.5$. Figure 8a differs with Figure 7b at upper layers, where probing errors for different representations don't have significant differences. Figure 8b is close to Figure 7b, also suggesting the algorithm selection mechanism. Figure 8c shows that the last layer encodes the information of all ridge predictors $\left\{\widehat{y}_i^{(j)}\right\}$, which is drastically different from the results in Figure 7c.

**Conjecture on two different algorithm selection mechanisms:** Based on the empirical findings, we conjecture two possible mechanisms of algorithm selection in transformer: (1) For small noise level data, transformers implement "concurrent-ICL algorithm selection", which means they concurrently implement ICL with algorithm selection. They wouldn't finish algorithms that are not have good performances. (2) For large noise level data, transformers implement "post-ICL algorithm selection", which means they would finish each algorithm. Then they select and output the best one. However, we need further experimental and theoretical work to inspect this conjecture.

## F  ABLATIONS

### F.1  SUPERVISED LEARNING WITH REPRESENTATION

**Probing results along training trajectory**  Figure 9a, Figure 9b, and Figure 9c show the probing error for $\Phi^\star(\mathbf{x}_i)$ at $\mathbf{x}$ and $y$ tokens and $\widehat{y}^{\Phi^\star \text{ridge}}$ at $\mathbf{x}$ tokens. As expected, all probe errors reduce through training steps, showing that the progress of learning $\Phi^\star$ is consistent with the progress of the training loss. At the 2000 training steps, transformer cannot recover the representation. At the 5000 training steps, the transformer starts memorizing the representation, starting showing differences

between lower and upper layers. From 5000 training steps to 10000, the trend of probe errors varying with layers remains the same.

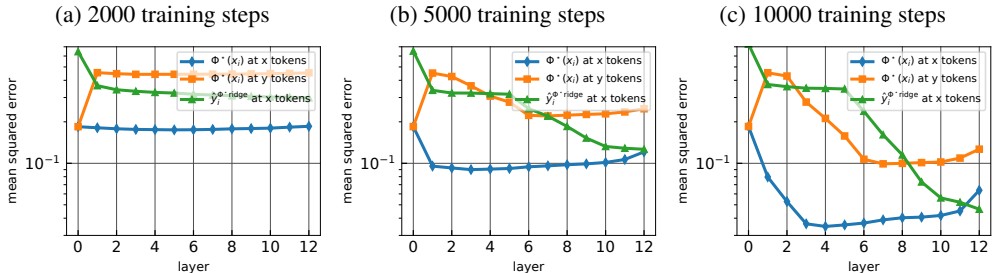

Figure 9: Probing errors for transformer trained after 2000, 5000, and 10000 steps. All three plots are for the training run on $(L, D, \sigma) = (2, 10, 0.1)$.

**Additional results for probing and pasting** Figure 10a plots the same probing errors as in Figure 3a with $(L, D, \sigma) = (3, 20, 0.1)$ (the green line there), except that we separate the errors of the first 4 tokens with the rest (token 5-41), but the probing training remains the same (pooled across all tokens). We observe that lower layers compute the representation in pretty much the same ways, though later layers forget the representations more for the beginning tokens (1-4) than the rest tokens.

Figure 10b plots the same pasting experiment as in Figure 4b, except that for noise level $\sigma = 0.5$ as opposed to $\sigma = 0.1$ therein. The message is mostly the same as in Figure 4b.

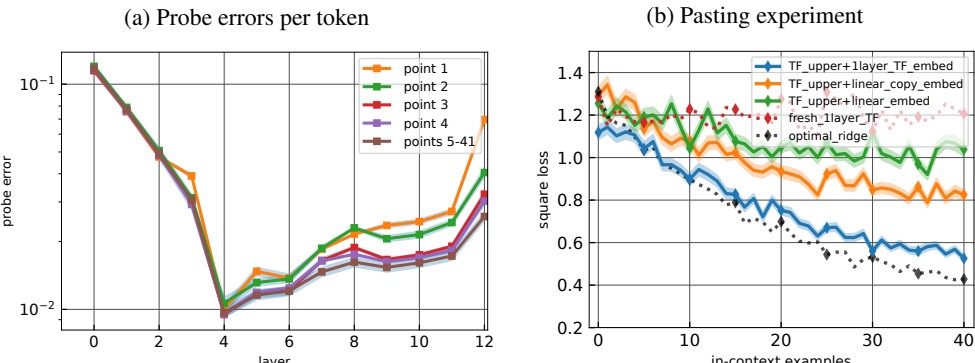

Figure 10: **(a)** Probing errors of $\Phi^\star(\mathbf{x}_i)$ in $\mathbf{x}_i$ tokens evaluated per-token. **(b)** Pasting results for the upper module of a trained transformer in setting $(L, D, \sigma) = (3, 20, 0.5)$.

## F.2 DYNAMICAL SYSTEMS

**Risk** Figure 11 gives ablation studies for the ICL risk in the dynamical systems setting in Section 4.2. In all settings, the trained transformer achieves nearly Bayes-optimal risk. Note that the noise appears to have a larger effect than the hidden dimension, or the number of input tokens.

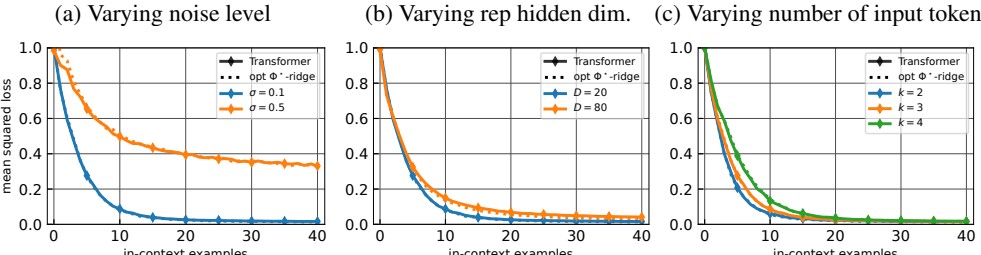

Figure 11: Ablation studies for the risk for Risk for fixed rep setting. Each plot modifies a single problem parameter from the base setting $(k, L, D, \sigma) = (3, 2, 20, 0.1)$.

**Probing** Figure 12a & 12b gives ablation studies for the probing errors in the dynamical systems setting in Section 4.2, with $D = 20$ instead of $D = 80$ as in Figure 5b & 5c. The message is largely similar except that in Figure 12a, all past inputs and intermediate steps in $\Phi^\star(\overline{\mathbf{x}}_i)$ are simultaneously best implemented after layer 4.

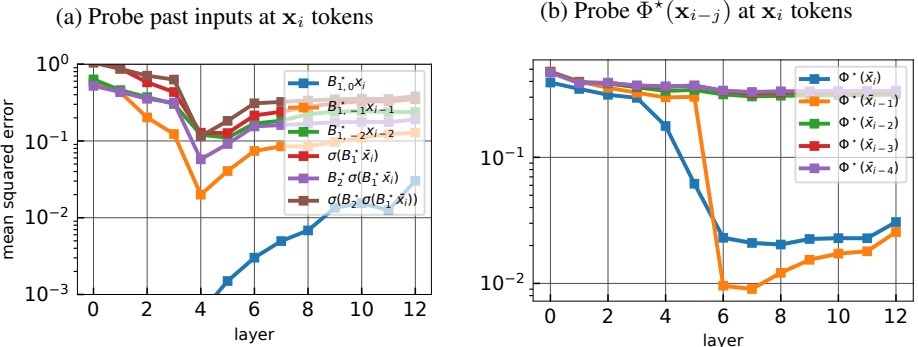

Figure 12: Ablation study for the probing errors in the dynamics setting. Here $(k, L, D, \sigma) = (3, 2, 20, 0.5)$, different from Figure 5 where $D = 80$.

### F.3 REPRESENTATION FUNCTIONS WITH HIGHER COMPLEXITY OR ALTERNATIVE STRUCTURES

Here we provide additional experiments when the representation function $\Phi^\star$ has a higher complexity or alternative structures (such as the nonlinearity $\sigma^\star$) from the ones in the main text.

We consider three scenarios for representations: 1. deep MLPs with $D = 20$, $L \in \{8, 12, 16\}$ and $\sigma = 0.1$; 2. wide MLPs with $D \in \{100, 200, 300\}$, $L = 20$ and $\sigma = 0.1$; 3. entry-wise nonlinear functions, where we replace MLP with entry-wise square, exponential, square leaky ReLU, and exponential leaky ReLU functions.

Figure 13 shows their risk curves. Both deep MLPs and other nonlinear functions match the optimal ridge regression. For wide MLPs, TF has slightly higher risk than optimal ridge regression. The risk curve of $D = 100$ is the lowest, with the other two on top of each other. Figure 14 shows probing error for deep MLPs. It seems that transformers would stop implement representations at the 7-th layer, with linear-ICL algorithm starts from 8-th layer.

Figure 15 shows the result of wide MLPs, we find that transformer needs one extra layer to get the representations if $D = 300$, which is wider than GPT2. Figure 16 shows for other entry-wise nonlinear functions. Transformer can get the representation within the first two layers, with copying happens in 4-th to 6-th layers. The gradient descent happens thereafter.

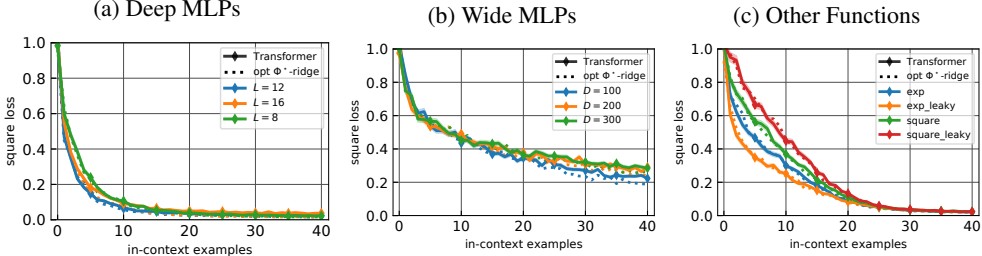

Figure 13: Test ICL risk for learning with complex representations. From left to right: deep MLPs, wide MLPs, nonlinear functions.

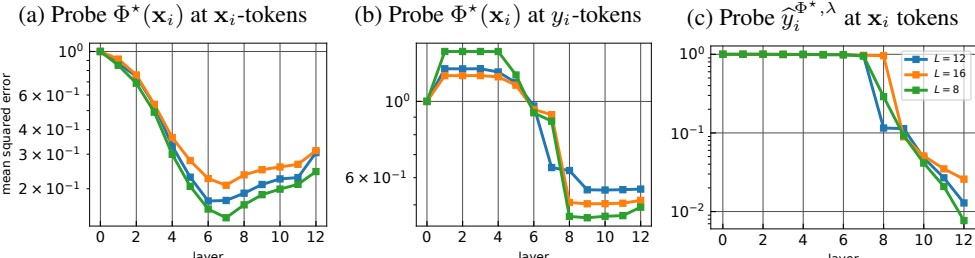

Figure 14: Probe $\Phi^\star(\mathbf{x}_i)$ and $\widehat{y}_i^{\Phi^\star,\lambda}$ at $\mathbf{x}_i$ and $y_i$ tokens with deep MLPs. We normalize the probing error, making the error at $0-$th layer to be 1.

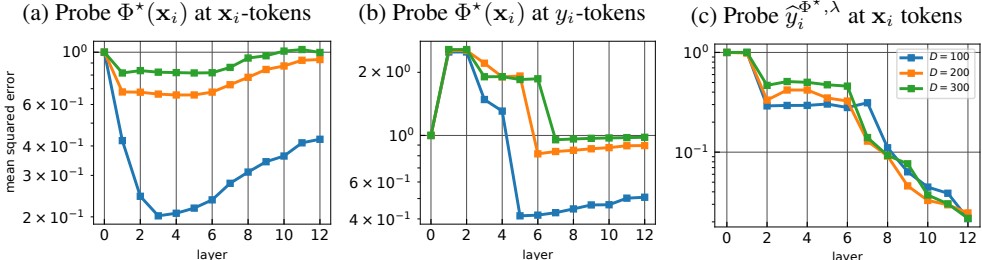

Figure 15: Probe $\Phi^\star(\mathbf{x}_i)$ and $\widehat{y}_i^{\Phi^\star,\lambda}$ at $\mathbf{x}_i$ and $y_i$ tokens with wide MLPs. We normalize the probing error, making the error at $0-$th layer to be 1.

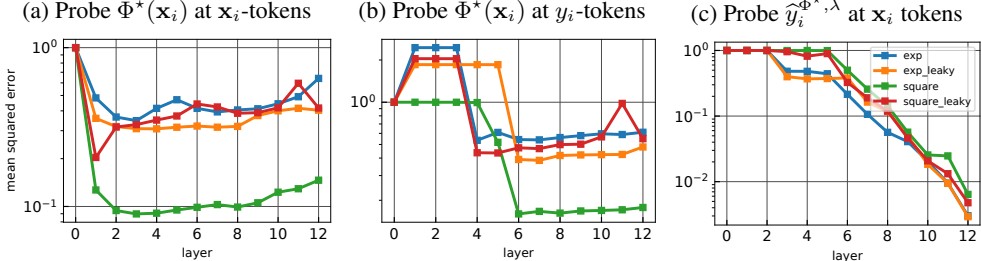

Figure 16: Probe $\Phi^\star(\mathbf{x}_i)$ and $\widehat{y}_i^{\Phi^\star,\lambda}$ at $\mathbf{x}_i$ and $y_i$ tokens with other non-linear functions. We normalize the probing error, making the error at $0-$th layer to be 1.

