# OpenReview forum: "How Do Transformers Learn In-Context Beyond Simple Functions? A Case Study on Learning with Representations"
_ICLR.cc/2024/Conference — ICLR 2024 poster_

### Official Review · Reviewer_z67a · 2023-10-30

**Soundness:** 3 good
**Presentation:** 4 excellent
**Contribution:** 3 good
**Rating:** 6
**Confidence:** 4

**Summary:**

This works studies in-context learning in transformers using synthetic data. It extends previous work, by studying composition of a *fixed* non-linear function (L-layer MLP) with a linear function that is learned in-context. This work provides a construction of a transformer that can solve this task, but also demonstrates it empirically on synthetic data. Additionally, the authors provide a mechanistic understanding of the algorithm implemented by a trained transformer.

**Strengths:**

The results extend the the setup of Garg et al. to study in-context learning with more complex function classes. In particular, transformers can learn a composition of a *fixed* non-linear function with a linear function learnt from context. The authors in Figure 1 provide evidence that a transformer matches the optimal predictor.

The mechanistic analysis is thorough, and provides compelling evidence for the underlying 3-step mechanism. It is surprising that the mechanism is consistent across multiple training runs and adds to the results of the paper.

The results also hold when multiple different non-linear representations are used which further strengthens the main claims of the paper. I would have liked to see the results of 4.1.1 more prominently in the main paper, but I understand the authors are constrained by space.

Overall, I think the results would be of interest to the community and the toy setup may be more representative of in-context learning in language models.

**Weaknesses:**

**Why are the constructive proofs important for understanding in-context learning in transformers?**
I am aware that there are prior works that design transformers that are capable of in-context learning. However, I am not convinced of the importance and significance of these results. Couldn't we also find weights for other architectures (like large MLPs or LSTMs) and argue that they are capable of in-context learning. Is the existence of these model weights informative of what is learnt in practice?

**Choice of non-linear functions.**  I think the authors could be more rigorous in evaluating the non-linear representations used in their setup. In particular, the non-linear representations are L-layer MLPs with the matrices being random orthogonal matrices. Do the results hold for other families of functions and does it fail to work for some other classes of functions? I think it would be helpful to clarify that the results are specific to this setup.

**Is the synthetic setup an accurate toy-model to understand language models?** Like previous work, all the results are on synthetic data. It remains unclear if the toy setup is representative of in-context learning in language models. What kind real world tasks are captured by a composition of a fixed non-linear function and a linear function that is learnt from context?

**Questions:**

1. Could the authors add more details on how the non-linear functions are created? Can the authors also clarify in the introduction/abstract that the functions are L-layer MLPs?

2. Is it possible to show some of these results on other families of non-linear functions? For example, what happens if the functions are polynomials or exponential functions of the input? Are there scenarios where it fails empirically?

3. What happens if we increase the number of layers used to create the representation from 5 to 15. Does the model start to fail if L=15 (even if transformer has only 11) layers or does it find a good approximation to the non-linear function using just 4-5 layers?

4. Results in appendix E were very interesting! As future work, it would be great to investigate how many different non-linear functions can be learnt. I would also be interested in understanding if in-context learning becomes difficult and if the model sometimes struggles to identify the right non-linear representation.

5. Ruiqi et al. (https://arxiv.org/abs/2306.09927) show that in-context learning fails if the linear functions are selected to be out-of-distribution. Is this also the case here?

---

> ### Author Response · Authors · 2023-11-20
> **Response to Reviewer z67a**
>
> We thank the reviewer for their valuable feedback. We respond to the specific questions as follows.
> > Why are the constructive proofs important for understanding in-context learning in transformers? I am aware that there are prior works that design transformers that are capable of in-context learning. However, I am not convinced of the importance and significance of these results. Couldn't we also find weights for other architectures (like large MLPs or LSTMs) and argue that they are capable of in-context learning. Is the existence of these model weights informative of what is learnt in practice?
>
> In our opinion, the importance for the constructive proofs is **not to give a yes/no answer** on “whether transformers can do this” or whether alternative architectures can do the same. Rather, the constructive proofs can
> * Demonstrate the **efficiency** of transformers for implementing these in-context learning procedures, i.e. only requiring a number of transformer layers and heads. This heavily uses the specific structure of the attention and MLP layers, instead of just invoking some universal approximation result;
> * Suggest concrete mechanisms (computing representation, copying, gradient descent) that can in turn guide the probing experiments.
> Alternative architectures such as LSTM or MLP (on concatenated input) may be able to implement a similar learning procedure, but **much less efficiently**, as it may be difficult to use them to efficiently replicate our mechanisms such as (batch) gradient descent or copying. They may be able to implement other mechanisms efficiently (such as online gradient descent for LSTMs), though.
>
> > Choice of non-linear functions… I think it would be helpful to clarify that the results are specific to this setup.
>
> > Could the authors add more details on how the non-linear functions are created? Can the authors also clarify in the introduction/abstract that the functions are L-layer MLPs?
>
> We mentioned “We instantiate the representation as shallow neural networks (MLPs)” in the paragraph before the list of contributions (Page 2). To clarify, we further highlighted “L-layer MLPs” in the abstract and the list of contributions in our revision. The choice of weight matrices and non-linear functions are described around Eq(4) in Section 4.1.
>
> > Is the synthetic setup an accurate toy-model to understand language models? Like previous work, all the results are on synthetic data. It remains unclear if the toy setup is representative of in-context learning in language models. What kind real world tasks are captured by a composition of a fixed non-linear function and a linear function that is learnt from context?
>
> While there is always a gap between such synthetic tasks and real tasks, we believe our fixed representation +  linear function is at least a more realistic model than prior works in this line, which primarily focuses on linear functions only. The fixed representation can model any “prior knowledge” that can be encoded in a representation (e.g. feature extractor), whereas the changing linear function can encode the specific classification task.
> As an illustrating toy example, consider the in-context learning problem
>
> "Give me the next word or phrase: ‘apple->means anger; ocean->means sadness; grass->means happiness; banana->means’”.
>
> The representation function for this problem would be to map each object to its color (apple -> red), and the linear function would be to map the color to an emotion (red -> anger). The feature map is fixed (relies on prior knowledge about the world) whereas the linear function needs to be learned in context (another such instance could involve a different map). Of course this is a toy example, but we believe its underlying structure (prior knowledge, then learn in context) could be broaderly present in real-world problems.

---

> > ### Author Response · Authors · 2023-11-20
> > **Response to Reviewer z67a**
> >
> > Due to space limit, we put the remaining responses here:
> >
> > > Is it possible to show some of these results on other families of non-linear functions? For example, what happens if the functions are polynomials or exponential functions of the input? Are there scenarios where it fails empirically?
> >
> > As suggested, we’ve tried entry-wise square, exponential, as well as {squared, exponentiated} leaky ReLU, directly on the input. These are equivalent to taking $\Phi^\star(x)=\sigma(Wx)$ with $W=I_d$, and thus is a special case of our MLP representation function with $L=1$ and these alternative activations.
> >
> > As shown in Figure 13(c) in our revision, trained transformers can again learn these maps fairly well. Probing also shows that lower layers calculate the representations, and upper layers implement linear ICL, similar as in our original settings (Figure 16).
> >
> >
> >
> > > What happens if we increase the number of layers used to create the representation from 5 to 15. Does the model start to fail if L=15 (even if transformer has only 11) layers or does it find a good approximation to the non-linear function using just 4-5 layers?
> >
> > We did some additional experiments with $L=8,12,16$ with our original architecture (12-layer transformer), see Figure 13(a), 14 in our revision. In these cases, trained transformers can still nearly match the Bayes-optimal algorithms (Figure 13). Further probing shows that trained transformers still appear to be implementing some approximate version of the representation function until the 7th layer, and then they start to implement ridge regression (Figure 14).
> >
> > We remark though that there may be some degeneracy problems when $L$ gets too large ( representation function itself may start to converge to some simpler function), thus we feel the results here are preliminary and worth further investigation.
> >
> >
> > > Ruiqi et al. (https://arxiv.org/abs/2306.09927) show that in-context learning fails if the linear functions are selected to be out-of-distribution. Is this also the case here?
> >
> > Ruiqi et al. consider two out-of-distribution tasks: OOD linear weight $w$, and OOD covariates $x$. In these OOD scenarios, we expect the same kind of failure here in our task: The theoretically constructed transformer can still implement the correct algorithm in these OOD scenarios, but the trained transformers (trained in in-distribution instances) may not perform well due to the failure of the empirical risk minimization on OOD.

---

### Official Review · Reviewer_R1v9 · 2023-11-01

**Soundness:** 3 good
**Presentation:** 3 good
**Contribution:** 4 excellent
**Rating:** 8
**Confidence:** 3

**Summary:**

The goal of this paper is to theoretically and empirically understand the mechanism of in-context learning with underlying representations. Specifically, the setting considered is where there is a fixed representation function, chosen to be an MLP, and the ICL problem is to learn ridge regression on these representations. The transformer must learn this fixed representation function during pretraining and a regression hypothesis in-context. The authors theoretically show that it is possible to construct transformers that can perform ridge regression in supervised and linear dynamical system settings on fixed representations. Empirically, the paper verifies that transformers can learn to perform this type of ICL by probing for the emergence of mechanisms and values that should emerge according to theoretical construction.

**Strengths:**

- The paper well-written.
- The experiments do a good job at validating the claims by probing for the relevant information.
- The results offer valuable insights into how in-context learning, which is very relevant and timely.

**Weaknesses:**

- Labels in figures and the figure captions can be more clear. For example, items like "TF_upper+1_layer_TF_embed" in Figure 4b are not very readable.
- Section 3 could be significantly condensed by considering theorem 2 as a generalization of theorem 1 instead of presenting them separately.
- Section 3.1 states that the representation function can be chosen arbitrarily, but Lemma B.3 requires a specific structure and non-linearity to work.
- Although the paper does a good job of illustrating the claimed mechanism, it does not analyze settings where the mechanism breaks.

**Questions:**

- What happens when the representation function is of a different form? If either the transformer does not have enough layers or the width, is there an approximate representation function learned on which regression is performed, or does the entire mechanism fall apart?
- How robust is learning of the representation function in settings where the pretraining data contains spurious correlations? Can we say anything about the transformer's ability to compositionally generalize with either the representation function, regression, or both?
- What is OLS in Fig 1b?

---

> ### Author Response · Authors · 2023-11-20
> **Response to Reviewer R1v9**
>
> We thank the reviewer for their valuable feedback. We respond to the specific questions as follows.
>
> > Although the paper does a good job of illustrating the claimed mechanism, it does not analyze settings where the mechanism breaks.
>
> > What happens when the representation function is of a different form? If either the transformer does not have enough layers or the width, is there an approximate representation function learned on which regression is performed, or does the entire mechanism fall apart?
>
> In theory, if the transformer does not have enough layers or width, then it can still implement an approximate algorithm (such as computing an approximate version of the representation function $\Phi^\star$, as the reviewer mentioned, combined with fewer gradient descent steps to implement approximate ridge regression).
>
> Empirically, we conducted some preliminary experiments with the same transformer architecture but more sophisticated representation functions, such as MLPs with higher depth or width. We’ve included these experiments in Appendix F.3 in our revision. Qualitatively, the trained transformer still exhibits similar mechanisms, as shown in the risk and probing curves (Figure 13-15), which gives evidence that they may indeed be implementing approximate versions of our claimed mechanism. Further questions such as what approximate representation function these are implementing could be an interesting question for future work.
>
>
> > How robust is learning of the representation function in settings where the pretraining data contains spurious correlations? Can we say anything about the transformer's ability to compositionally generalize with either the representation function, regression, or both?
>
> By “spurious correlations”, did the reviewer mean OOD-like scenarios where the test time representation function is different from the training time one in function space, but they happen to be identical on the training data distribution? In that case, we can still guarantee the existence of a transformer that can do well on both the test time representation function and the training data, but we won’t have a statistical guarantee on how training based on empirical risk minimization can actually find this transformer, as the statistical guarantee (e.g. the one sketched in the last paragraph of Section 4.1) only holds in-distribution.
>
> > Section 3.1 states that the representation function can be chosen arbitrarily, but Lemma B.3 requires a specific structure and non-linearity to work.
>
> Did the reviewer mean Section 4.1? Right after “The representation function $\Phi^\star$ can in principle be chosen arbitrarily” in Section 4.1, we stated “As a canonical and flexible choice for both our theory and experiments, we choose $\Phi^\star$ to be a standard L-layer MLP”, followed by Eq(4) which describes the MLP architecture in details (including the choice of non-linearity). Lemma B.3 follows the specifications made here.
>
> > Labels in figures and the figure captions can be more clear. For example, items like "TF_upper+1_layer_TF_embed" in Figure 4b are not very readable.
>
> Thank you for pointing this out. We have added a short explanation about this in the caption of Figure 4 in our revision.
>
> > What is OLS in Fig 1b?
>
> The OLS means ordinary least squares (i.e. standard linear regression) on top of the representation function $\Phi^\star$.

---

> > ### Comment · Reviewer_R1v9 · 2023-11-21
> > **Keeping my score**
> >
> > Thank you for your responses!
> >
> > Thank you for your additions in Appendix F.3, that is very helpful.
> >
> > > By “spurious correlations”, did the reviewer mean OOD-like scenarios where the test time representation function is different from the training time one in function space, but they happen to be identical on the training data distribution?
> >
> > I did not mean that your test time representation function is different. Rather, the data is sampled differently at test time. Right now, your x and z are sampled from a Gaussian prior. What if the covariance in these Gaussian priors is not diagonal? There is often sampling bias in real world data, containing correlations that are not causal. If you train with such correlations present in the data, but show balanced data at test time for ICL, does the model reveal the true representation function?
> >
> > > Did the reviewer mean Section 4.1?
> >
> > Sorry, yes. I understand that you choose L-layer MLP, but I was questioning the statement "The representation function can in principle be chosen arbitrarily". But I see that other representation functions are addressed now in Appendix F.3.
> >
> > I'm going to keep my score.

---

> > > ### Author Response · Authors · 2023-11-22
> > > **Response**
> > >
> > > Thank you for your response and the support on our paper!
> > >
> > > Re the OOD question: Yes, if the transformer is trained with (non-diagnoal) Gaussian prior for the x, then we expect it to utilize this structure and it may perform suboptimally when test-time x is say spherical Gaussian. It would also be unclear whether it "reveals the true representation function".

---

### Official Review · Reviewer_67Bi · 2023-11-01

**Soundness:** 2 fair
**Presentation:** 3 good
**Contribution:** 3 good
**Rating:** 6
**Confidence:** 4

**Summary:**

Results illustrating the performance of transformers in ICL tasks that necessitate some degree of representation learning are presented. The theory can be partially validated through probing experiments.

**Strengths:**

Clear and well-written.

This paper is one of the pioneering efforts to formalize how transformers execute ICL tasks that necessitate a degree of representation learning.

**Weaknesses:**

The theory only encompasses representational results by providing some settings of the parameters in a way that a transformer performs an ICL task. Given the transformer's highly expressive capability, these types of constructions are generally relatively straightforward.

There's no assurance that these theoretical constructs are truly internalized by the model during the training process. Although probing experiments gave us some confidence that, in specific instances, the theory can predict the model's behavior, these types of experiments generally don't offer robust guarantees. As a result, while the theory is logical and sometimes mirrors empirical events, it could be counterproductive to lean too heavily on these theoretical constructs. It might be necessary to carry out an analysis of training dynamics in order to theoretically determine under which conditions the model actually aligns with the theoretical constructs.

**Questions:**

The reviewer is open to learning about new evidences or analyses which address the points of the “Weaknesses” section above in this review.

---

> ### Author Response · Authors · 2023-11-20
> **Response to Reviewer 67Bi**
>
> We thank the reviewer for their valuable feedback. We respond to the specific questions as follows.
> > The theory only encompasses representational results by providing some settings of the parameters in a way that a transformer performs an ICL task. Given the transformer's highly expressive capability, these types of constructions are generally relatively straightforward.
>
> While transformers are known to be highly expressive approximators of sequence-to-sequence functions, the **efficiency** and the **internal mechanisms** of the transformer still matters a lot. Our transformer constructions in Theorem 1&2 are efficient (transformer has low number of layers and heads). Further, both its high-level structure (such as computing representation + using gradient descent to do linear regression) + lower-level operations such as copying show up experimentally in trained transformers via probing. We believe these make the construction itself much more interesting and reflective of reality than a simple universal approximation type argument.
>
> > There's no assurance that these theoretical constructs are truly internalized by the model during the training process. Although probing experiments gave us some confidence that, in specific instances, the theory can predict the model's behavior, these types of experiments generally don't offer robust guarantees. As a result, while the theory is logical and sometimes mirrors empirical events, it could be counterproductive to lean too heavily on these theoretical constructs.
>
> We appreciate this thoughtful question. First of all, we agree that theoretical construct is just one way to guide the probing experiments, there may be other ways to formalize hypotheses about the actual mechanism, and we don’t necessarily need to rely solely on the theory.
> Nevertheless, we still believe that the probing results on their own are actually a strength of our work, in that we are the first to give evidence of these mechanisms (especially lower-level ones such as copying) for in-context learning on trained transformers. In addition to probing, we also conducted a “pasting” experiment (Figure 4) as a more controlled way to test the linear ICL capability of the upper module of the trained transformer.
> We believe further probing studies, potentially going beyond the theoretical constructs, is an important direction which we would like to leave as future work.
>
> > It might be necessary to carry out an analysis of training dynamics in order to theoretically determine under which conditions the model actually aligns with the theoretical constructs.
>
> We agree that the training dynamics is an interesting future direction as well. In our ablations we already had some preliminary experiments on the training trajectory (Appendix F.1, Figure 9) where we found that the representation module and the linear ICL module gets simultaneously learned in the early training stage. Further studies in this direction would be interesting.

---

> > ### Comment · Reviewer_67Bi · 2023-11-20
> > **Thank you for the response**
> >
> > Thank you for the clarifications!
> >
> > My current evaluation is still close to a score of 6 (marginal accept): despite the limitations (that the theory does not guarantee in what settings do the construction can be learned, and the experimental results in this synthetic setting is mostly expected), the theoretical community should still be able to learn from reading this paper because the theory establishes an interesting setting and is well-written.

---

> > > ### Author Response · Authors · 2023-11-22
> > > **Response**
> > >
> > > Thank you for the response and the support on our paper!

---

### Official Review · Reviewer_oQN2 · 2023-11-03

**Soundness:** 3 good
**Presentation:** 3 good
**Contribution:** 2 fair
**Rating:** 6
**Confidence:** 3

**Summary:**

This paper focuses on understanding the internal mechanism by which a Transformer model solves an in-context learning task where the label $y$ for an instance $x$ linearly depends on a representation $\phi^{\star}(x)$.  A recent line of work has focused on explicitly constructing transformer models that can simulate various learning methods (e.g., gradient descent) on a training objective defined by the in-context labeled examples during a forward pass of the Transformer model. This paper extends this line of work by considering a more general data model where the final label depends on the input instance through a linear function of a representation. The paper provides explicit constructions for Transformer networks that can simulate ridge regression for 1) supervised learning with a representation, and 2) learning dynamical systems with a representation. The explicit constructions first aim to employ the underlying representation map $\phi^{\star}$ in the lower layers of the Transformer and then implement gradient descent in the upper layers of the Transformer.

Through experiments on synthetic datasets, the paper demonstrates that the performance of in-context learning via Transformers closely agrees with the performance of an optimal ridge predictor. Through probing analysis of the Transformer models, the authors show evidence that supports representation mapping following by label prediction aspects of their constructions.

**Strengths:**

1) The paper successfully extends the recent line of work on showing the feasibility of in-context learning via empirical risk minimization during a forward pass by considering data models where labels depend on the input feature via a representation map.
2) The paper is well-written and explains the key contributions and techniques clearly.
3) The empirical results (on synthetic datasets) do indicate the feasibility/presence of the explicit in-context learning mechanism hypothesized in the paper.

**Weaknesses:**

1) Novelty of the technical contributions is limited given prior works of similar flavor that provide the feasibility of empirical risk minimization during forward pass. One of the aspects which authors claim to be novel is that they allow for representation-based learning. However, the underlying assumption is that the representation map is a multi-layer MLP model, which Transformers should be easily able to simulate through its MLP layers. In that sense, the results in not very surprising.

2) The probing analysis is done only on a synthetic setup. It would be nice to get some supporting evidence for the proposed in-context learning mechanism on a real dataset.

Minor issues:

1) The authors may want to formally introduce/discuss the pre-training phase (e.g., Eq (10)) which learns the representation map early in the section on preliminaries.

2) In the paragraph on **In-context learning** in Section 2, $\mathcal{D}^{(j)}$ and $\mathbf{w}_{\star}^{(j)}$ are not defined before their usage.

**Questions:**

1) In general, the transformers are known to be universal approximators. In light of these, could authors comment on the significance of the key contributions of the paper, i.e., learning a representation map before applying gradient-descent in the representation space?

2) Could the authors elaborate on using a **linear model** for their investigation on the upper module via pasting (Figure 4)?

3) Currently the non-linearity in the true representation map is closely tied to the nonlinearity used in the Transformer network. Could the authors comment on generalizing this to broader nonlinearities in the true representation map? How would it increase the required number of layers?

---

> ### Author Response · Authors · 2023-11-20
> **Response to Reviewer oQN2**
>
> We thank the reviewer for their valuable feedback. We respond to the specific questions as follows.
>
> > Novelty of the technical contributions is limited... However, the underlying assumption is that the representation map is a multi-layer MLP model, which Transformers should be easily able to simulate through its MLP layers. In that sense, the results in not very surprising… could authors comment on the significance of the key contributions of the paper?
>
> For the MLP part of the theory, we agree the high-level idea itself is not very surprising, as mentioned. However, instantiating this idea into a full rigorous construction involves several other new ingredients different from existing work. For example, we required an efficient implementation of N parallel gradient descent simultaneously for *predicting at every token* using a decoder transformer (Proposition B.4), whereas similar existing work (Bai et al. 2023, Zhang et al. 2023a) only implements a single gradient descent for predicting at the last token only using an encoder transformer. See the “Proof techniques” paragraph after Theorem 1 for more details.
> Further, another core contribution of this paper lies in using theory to guide empirical mechanistic study in a fine-grained (low-level) way, and indeed validating that the mechanisms identified from the theory show up in trained transformers (Figure 3-5). Along with the findings, we also proposed new techniques such as pasting (Figure 4) which may be of further interest.
>
> > The probing analysis is done only on a synthetic setup. It would be nice to get some supporting evidence for the proposed in-context learning mechanism on a real dataset.
>
> While our setting is synthetic, the setting (linear function on top of common representation) is arguably more realistic than most existing works in this direction that focus on linear functions. The synthetic setting also allows probing analyses to be done in conjunction with rigorous theoretical studies. Nevertheless, We agree that some probing analyses on e.g. real data or even more realistic synthetic data is an important question, which we believe is out of the scope of this paper but an interesting direction for future work.
>
>
> > In general, the transformers are known to be universal approximators. In light of these, could authors comment on the significance of the key contributions of the paper, i.e., learning a representation map before applying gradient-descent in the representation space?
>
> While the high-level plan of “learning representation + applying gradient descent” can be done by universal approximation, and this paradigm is not so surprising given prior work,**how** we instantiate this construction is important and makes a difference. The way we instantiate it in our Theorem 1&2 are efficient (transformer has low number of layers and heads), and lower-level operations such as copying we use in our construction do show up experimentally via probing. We believe these make the construction itself much more interesting and reflective of reality than a simple universal approximation type argument.
>
> > Could the authors comment on generalizing this to broader nonlinearities in the true representation map? How would it increase the required number of layers?
>
> If the true representation map involves other nonlinearities beyond leaky ReLU, we could approximate such a non-linearity by a linear combination of ReLU functions. Therefore we can construct a similar transformer with potentially more heads within each layer (for approximating scalar nonlinearity using a linear combination of ReLUs), but the same number of layers.
>
> > Could the authors elaborate on using **a linear model for their investigation** on the upper module via pasting (Figure 4)?
>
> Is the question about why we **use a linear model as the data distribution** to investigate the upper module via pasting? Our goal is to test if the upper layers are simply implementing linear regression, or if they are implementing a more complex algorithm like linear regression with feature learning. The linear model is a sensible choice for testing this. In our experiment, we generated data using a linear model and passed it through the upper layers. The results suggest the upper layers are just doing vanilla linear regression.
>
> > In the paragraph on In-context learning in Section 2, $\mathcal{D}^{(j)}$ and $\mathbb{w}_{\star}^{(j)}$
>  are not defined before their usage.
>
> Thanks for spotting these issues. We’ve updated them accordingly in our revision.

---

> > ### Comment · Reviewer_oQN2 · 2023-11-22
> > **Adjusted the score**
> >
> > Thank you for your response.
> >
> > Some of my concerns have been addressed by your response. I have updated my score to 6.

---

### Author Response · Authors · 2023-11-20
**Revision uploaded & clarification on contributions**

We thank all reviewers again for their valuable feedback on our paper. We have incorporated the reviewers’ suggestions and uploaded a revised version of our paper. We have also added more experiments in Appendix F.3. For clarity, all changes (other than typo fixes) are marked in blue.

Additionally, since several reviewers asked, here we make **a few clarifications regarding the contributions of our work**:
* The **pasting** experiment (Figure 4, Page 8) is one of our main empirical results, which did not seem to be noticed by most reviewers. By testing the upper module independently, it gives evidence that the upper module is indeed implementing linear ICL on its own. The pasting technique may also be of further interest.
* Regarding the contributions of our **constructive proofs**, we believe its importance is **not to give a yes/no answer** on whether transformers can implement a good learning algorithm, but rather to
  - Show that transformers can implement it **efficiently** (with a small number of layers and heads), rather than invoking some universal approximation result, and
  - Suggest concrete mechanisms that can guide the probing experiments.
We believe the combination of the size bounds, the identified mechanisms, and the experimental findings (trained transformers indeed mirror them) justify the necessity and contributions of our rigorous transformer constructions.

---

### Meta-Review · Area_Chair_yocF · 2023-12-13

**Metareview:**

The paper studies in-context learning for slightly more 'complex' problems than have previously been studied. In particular, rather than simply looking at linear functions etc., they look at linear functions of a representation of the input (which may vary). There are some questions regarding whether the theoretical results say much about actual in-context learning. Nevertheless the reviewers were all mostly positive and of the view that the paper makes some progress towards understanding in-context learning.

**Justification For Why Not Higher Score:**

Theory is inadequate to explain experimental behaviour. Overall, slightly borderline paper, but above the threshold.

**Justification For Why Not Lower Score:**

Interesting paper.

---

### Decision · Program_Chairs · 2024-01-16

Accept (poster)